# Atomically defined angstrom-scale all-carbon junctions

Zhibing Tan[1,4], Dan Zhang[1,4], Han-Rui Tian[1,4], Qingqing Wu[2,4], Songjun Hou[2], Jiuchan Pi[1], Hatef Sadeghi [2], Zheng Tang[1], Yang Yang[1], Junyang Liu[1], Yuan-Zhi Tan [1], Zhao-Bin Chen[1], Jia Shi[1], Zongyuan Xiao[1,3], Colin Lambert [2], Su-Yuan Xie[1] & Wenjing Hong [1,3]

Full-carbon electronics at the scale of several angstroms is an expeimental challenge, which could be overcome by exploiting the versatility of carbon allotropes. Here, we investigate charge transport through graphene/single-fullerene/graphene hybrid junctions using a single-molecule manipulation technique. Such sub-nanoscale electronic junctions can be tuned by band gap engineering as exemplified by various pristine fullerenes such as $C_{60}$, $C_{70}$, $C_{76}$ and $C_{90}$. In addition, we demonstrate further control of charge transport by breaking the conjugation of their $\pi$ systems which lowers their conductance, and via heteroatom doping of fullerene, which introduces transport resonances and increase their conductance. Supported by our combined density functional theory (DFT) calculations, a promising future of tunable full-carbon electronics based on numerous sub-nanoscale fullerenes in the large family of carbon allotropes is anticipated.

[1] State Key Laboratory of Physical Chemistry of Solid Surfaces, iChEM, NEL, College of Chemistry and Chemical Engineering, Xiamen University, 361005 Xiamen, China. [2] Department of Physics, Lancaster University, Lancaster LA1 4YB, UK. [3] Graphene Industry and Engineering Research Institute, Xiamen University, 361005 Xiamen, China. [4] These authors contributed equally: Zhibing Tan, Dan Zhang, Han-Rui Tian, Qingqing Wu. Correspondence and requests for materials should be addressed to Z.X. (email: xiaozy@xmu.edu.cn) or to C.L. (email: c.lambert@lancaster.ac.uk) or to S.-Y.X. (email: syxie@xmu.edu.cn) or to W.H. (email: whong@xmu.edu.cn)

The high operational speed of all-carbon electronics[1,2], employing one-dimensional (1D) carbon nanotubes (CNTs) as transport materials and two-dimensional (2D) graphene as electrodes, holds significant promise beyond silicon electronics[3,4]. Although the CNTs and carbon nanoribbons[5] have been in the fabrication of computing circuits[6], the ability of CNTs to conduct charge is sensitive to atomic structural topology as well as the presence of edge defects that are hard to control due to structural uncertainties and unavoidable dangling bonds in the low-dimensional carbon materials[7,8]. Therefore, there is an urgent need to develop the potential of defect-free zero-dimensional (0D) fullerenes and atomically defined carbon allotropes for carrying out band gap engineering and constructing all-carbon electronics at angstrom scale based on the well-established fullerene science[9,10].

Early pioneering single-$C_{60}$ junctions were fabricated by contacting fullerenes with a scanning tunneling microscope (STM) or using electron-beam lithography to gold electrodes[11–13]. Recent developments suggest that fullerene-based junctions could exhibit a pronounced Seebeck effect[14–16] and that charge transport could be precisely manipulated depending on the contact area between the molecule and electrodes[17], for example, through chemical modification of the fullerene with planar anchor groups such as pyrenes, which bind to the graphene via van der Waals interactions[16]. An even simpler strategy involves the bridging of graphene electrodes via the overlap of graphene and fullerene π orbitals, which could be achieved using a single-molecule junction techniques[18–22], whose electrode gap can be controlled with angstrom precision.

Here we fabricate and investigate charge transport through a series of hybrid graphene/single-fullerene/graphene junctions with pristine fullerene cores using mechanically controlled break junction (MCBJ) technology[18]. It is found that the fullerene can be bridged between two electrodes of the chemical vapor deposition (CVD) graphene via van der Waals interaction to form the single fullerene junctions. More importantly, the conductance of the graphene/single-fullerene/graphene junctions can be controlled by inserting different fullerenes with a variety of energy gaps. The charge transport can be further tuned through heteroatom doping, which introduces desirable transport resonances. These results demonstrate the exquisite controllability of all-carbon electronics via band gap engineering.

## Results

**Charge transport of all-carbon junctions.** As shown in Fig. 1a, several typical fullerenes, such as $C_{60}$, $C_{70}$, $C_{76}$, and $C_{90}$ are synthesized using our low-pressure benzene–oxygen diffusion combustion approach (See "Methods" for more details), and charge transport through the graphene/single-fullerene/graphene junctions are investigated using the MCBJ technique with CVD graphene-coated copper wire as electrodes.

During the dynamic opening and closing process, a nanogap is created between the two graphene electrodes and increased until the disappearance of the tunneling current (See "Methods" for more details on MCBJ technique and graphene electrodes). In the presence of fullerenes in the solution, a fullerene molecule can be trapped between the graphene electrodes via van der Waals interactions to form a graphene/single-fullerene/graphene junction, which is further connected with the external circuit with 100 mV bias voltage to monitor the current across the junction. As demonstrated in Fig. 1b, the typical individual conductance–displacement curves of $C_{60}$ show clear plateaus around $2.0 \times 10^{-5}$ $G_0$ ($G_0$ is the quantum conductance that equals $2e^2/h$), which represent the conductance of single-$C_{60}$ junctions bridged between the two graphene electrodes, while there is only the exponential decay of tunneling in the absence of fullerenes. Furthermore, it is found that the molecular

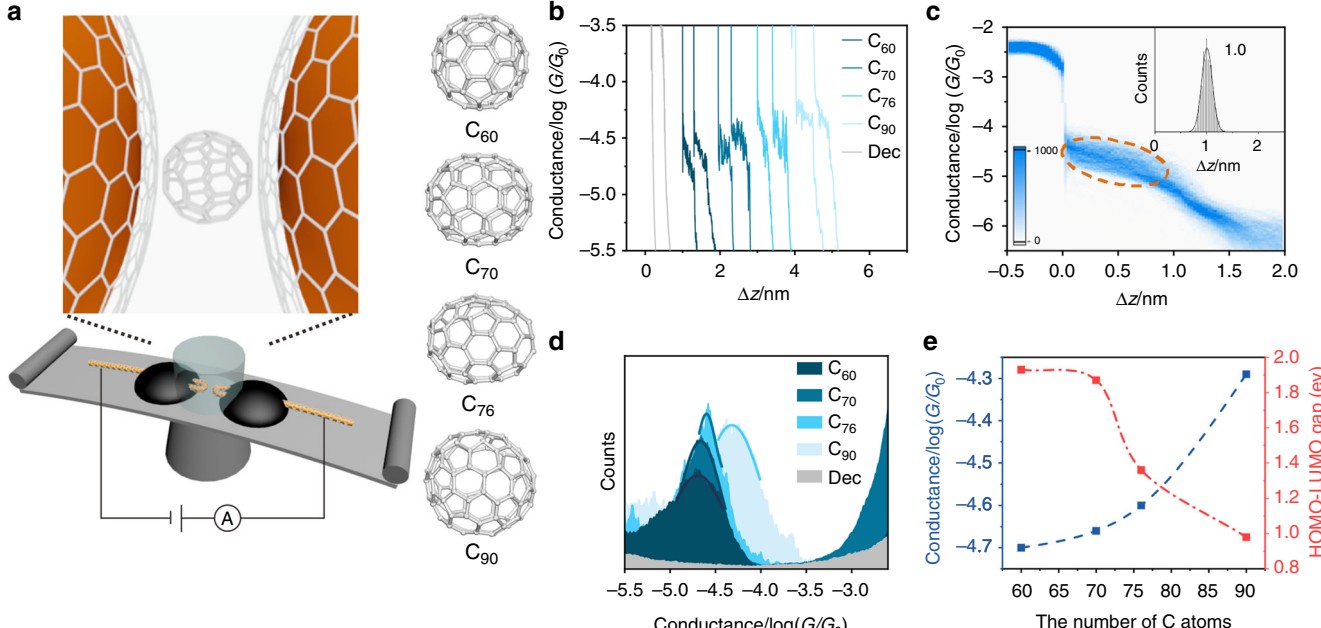

**Fig. 1** Conductance measurement of graphene/single-fullerene/graphene junctions. **a** Schematic of mechanically controlled break junction and graphene/single-fullerene/graphene junction and chemical structure of $C_{60}$, $C_{70}$, $C_{76}$, and $C_{90}$. **b** Typical individual conductance–displacement curves for graphene/blank or single-$C_{60}$, $C_{70}$, $C_{76}$, and $C_{90}$/graphene junctions. **c** Two-dimensional conductance histogram of single-$C_{60}$ junction obtained from ~1700 traces, the high trigger was set as $3.2 \times 10^{-3}$ $G_0$ ($G_0$ is the quantum conductance that equals $2e^2/h$.) to avoid the damage of the graphene electrodes. The top right inset is the relative displacement distribution ranging from $3.2 \times 10^{-4}$ to $4.0 \times 10^{-6}$ $G_0$. **d** One-dimensional conductance histogram of graphene/single-$C_{60}$, $C_{70}$, $C_{76}$, $C_{90}$/graphene junctions and blank experiments. **e** Conductance of the graphene/single-fullerene/graphene junctions and the highest occupied molecular orbital–lowest unoccupied molecular orbital gap values of fullerenes vs. the number of carbon atoms in fullerenes

plateaus shift to higher conductances as the sizes of the fullerenes increase, such that the conductance of $C_{90}$ is almost half an order of magnitude higher than that of $C_{60}$. To further investigate the conductance variations among single-$C_{60}$ junctions, the corresponding 2D histogram of the graphene/single-$C_{60}$/graphene measurement (Fig. 1c) constructed from ~1700 individual conductance–displacement curves display a well-defined conductance intensity cloud, suggesting that the fullerene junctions possess well-defined charge transport properties (the 2D conductance histograms of graphene/blank or single-$C_{70}$, $C_{76}$, and $C_{90}$/graphene junctions are shown in Supplementary Figs. 5 and 6). The relative displacement distributions of conductance plateaus (The upright inset of Fig. 1c.) indicates that most of the $C_{60}$ junctions are ~1 nm, which is consistent with the size of the fullerene with additional van der Waals distance with the graphene electrodes.

To enable a quantitative comparison, we construct the 1D conductance histograms shown in Fig. 1d, which indicate that the most probable conductance peaks of graphene/single-$C_{60}$, $C_{70}$, $C_{76}$, and $C_{90}$/graphene are located at $2.0 \times 10^{-5}$ $G_0$, $2.2 \times 10^{-5}$ $G_0$, $2.5 \times 10^{-5}$ $G_0$, and $5.1 \times 10^{-5}$ $G_0$ respectively, demonstrating that there is a >250% conductance enhancement from $C_{60}$ to $C_{90}$. This result shows that the conductance of the single-$C_{60}$ junctions with graphene electrodes is lower than that of gold electrodes[23–25], which agrees with the previous report that the π–π interaction is enough to construct molecular junctions, but the conductance is reduced[26]. Data of the closing process of graphene/single-$C_{60}$/graphene junctions are given in Supplementary Fig. 7, which indicates that closing process form molecular junctions with the same length the opening process, but the conductance ($5.0 \times 10^{-5}$ $G_0$) is slightly larger than that of the opening process because of the contribution of pure tunneling[27]. As shown in Fig. 1e, the conductance of the graphene/single-fullerene/graphene junctions increases with the size of fullerenes from $C_{60}$ with ~7 angstroms to $C_{90}$ with ~9 angstroms. We also analyzed the most probable displacement of junctions with different sizes of fullerene (Supplementary Fig. 6), and found that the increased length correlates with fullerene size, which further validate the trapping of single fullerene between the two graphene electrodes. Further investigation suggests that the increase is correlated with the decrease of the fullerene HOMO (highest occupied molecular orbital)–LUMO (lowest unoccupied molecular orbital) gap (The HOMO–LUMO gaps of fullerenes were analyzed from the ultraviolet–visible (UV-Vis) spectra, see Supplementary Table S1 in detail.), which overcomes the tendency for electrical conductance to decrease with molecular length[28–30]. This suggests

that the conductance of graphene/single-fullerene/graphene junctions could be fine-tuned by making further choices from the huge family of pristine fullerenes with structure-dependent band gaps. An illustration of the competing relation between the band gap and molecular size is provided in recent studies of the length-dependent conductance of porphyrin wires formed from oligoyne-linked porphyrin units or from fused porphyrin tapes[31,32]. In both cases, their HOMO–LUMO gaps shrink with increasing length, but for oligoyne-linked wires, this decrease is rather slow and therefore their conductance decreases with length. In contrast, the HOMO–LUMO gap of the more strongly coupled porphyrin tapes decreases more rapidly with length and their conductance are found to increase with length.

**Control of charge transport via conjugation tuning**. To further explore the effect of fullerene structure on charge transport through graphene/single-fullerene/graphene junctions, we investigate the conductance of the low-conjugated hydrofullerene $C_{50}H_{10}$, whose π system is interrupted by ten hydrogen-bonded carbon atoms (Structure shown in Fig. 2a.). We measure the single-molecule conductance of $C_{50}H_{10}$ with graphene electrodes (Fig. 2b for the 2D conductance histogram), and find that the displacement distribution shown in the inset of Fig. 2b are shorter than that of $C_{60}$. This suggests that the $C_{50}H_{10}$ molecule is aligned with the short axis in the junctions and the graphene electrodes are contacted with the conjugated part of the $C_{50}H_{10}$ molecule. The conductance of the low-conjugated graphene/single-$C_{50}H_{10}$/graphene junctions is located at $2.2 \times 10^{-6}$ $G_0$, which is almost one order of magnitude lower than that of the conjugated single-$C_{60}$ (Fig. 2c). These results show that, for graphene/single-fullerene/graphene junctions, the conductance decreases significantly when the conjugation of the π system is interrupted.

**Control of charge transport via heteroatom doping**. Heterofullerene is expected to have applications in the field of photoelectric devices and superconductivity, because the introduction of nitrogen heteroatoms into the fullerene cage leads to significant perturbations of the geometric and electronic character of the fullerene clusters[33]. We synthesize the $D_{2h}$ symmetric $C_{120}$ fullerene dimer directly linked by cyclic $C_4$ units in a [2+2] cycloaddition and nitrogen-doped fullerene dimers $(C_{59}N)_2$ with the same atomic numbers (See the Methods section). Figure 3e shows that the most probable conductance of graphene/single-$C_{120}$/graphene junctions is $1.3 \times 10^{-5}$ $G_0$ (Fig. 3c for the 2D conductance histogram). Our previous studies on dumbbell

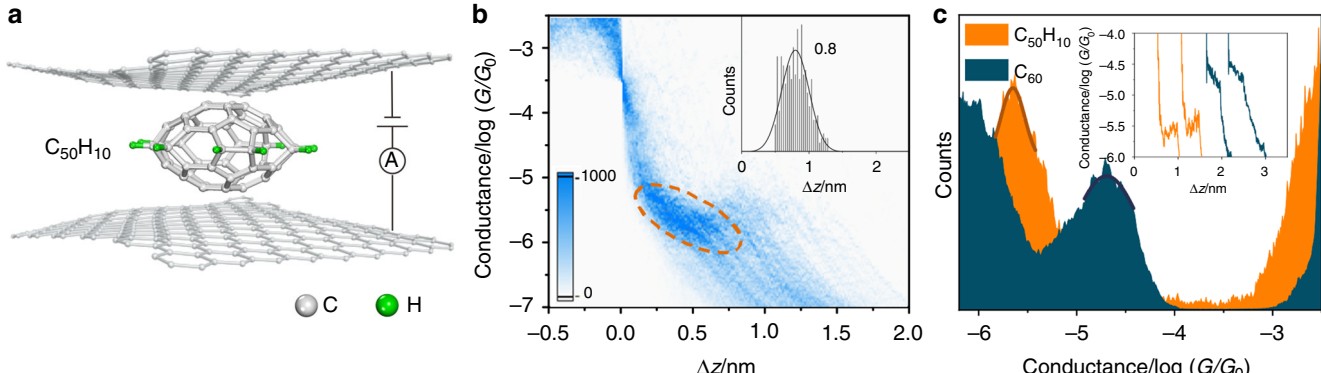

**Fig. 2** Charge transport through graphene/single-$C_{50}H_{10}$/graphene junctions. **a** Schematic of graphene/single-$C_{50}H_{10}$/graphene junctions. **b** Corresponding two-dimensional conductance histogram obtained from ~1000 traces. The top right inset is the relative displacement distribution (The conductance range to determine the displacement are from $3.2 \times 10^{-4}$ to $6.3 \times 10^{-7}$ $G_0$.). **c** Comparison of one-dimensional conductance histogram of the graphene/single-$C_{50}H_{10}$/graphene junctions. The top right inset is the individual conductance–displacement curves

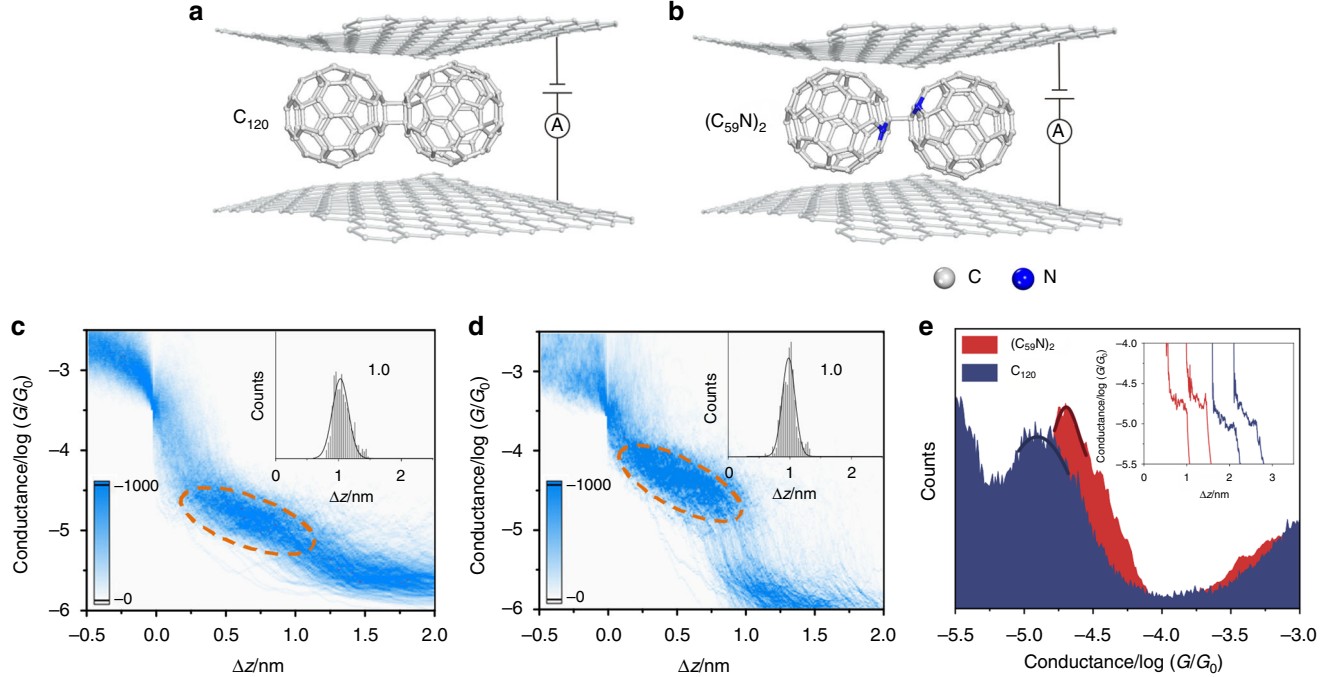

**Fig. 3** Heteroatoms doping in graphene/single-fullerene/graphene junctions. **a**, **b** Schematic of graphene/single-$C_{120}$ (**a**) and $(C_{59}N)_2$ (**b**)/graphene junctions. **c**, **d** Corresponding two-dimensional conductance histogram obtained from ~1000 traces of $C_{120}$ (**c**) and $(C_{59}N)_2$ (**d**). The top right inset is the relative displacement distribution (The conductance range to determine the displacement of $C_{120}$ and $(C_{59}N)_2$ are from $3.2 \times 10^{-4}$ to $4.0 \times 10^{-6}$ $G_0$, $1.0 \times 10^{-5}$ $G_0$, respectively.). **e** Ono-dimensional conductance histogram of the graphene/single-fullerene/graphene junctions composed of $C_{120}$ and $(C_{59}N)_2$. The top right inset is the individual conductance–displacement curves

fullerene compounds reveal that charge transport through the buckminsterfullerene anchoring group is orders of magnitude higher than through the fully extended molecule[34]. Since the measured conductance values and the conductance plateaus of $C_{120}$ (Fig. 3c inset for the conductance plateau distribution) are similar to those of $C_{60}$, this suggests that $C_{120}$ lies on the graphene electrodes as shown in Fig. 3a. Compared with the conductance of graphene/single-$C_{120}$/graphene junctions, the conductance of graphene/single-$(C_{59}N)_2$/graphene (Fig. 3d for the 2D conductance histogram and conductance plateau distribution) increases by 150% and reaches $2.1 \times 10^{-5}$ $G_0$ (Fig. 3e), demonstrating that nitrogen atom doping enhances charge transport through graphene/fullerene/graphene junctions.

**Density functional theory (DFT) simulations**. To elucidate the conductance evolution of the graphene/single-fullerene/graphene junctions, their cross-plane conductance values are calculated with the combination of the ab initio DFT package SIESTA[35] and the quantum transport code GOLLUM[36] (see computational details in the following "Methods" section). Figure 4a depicts the four-terminal model, where a single target molecule is sandwiched between two parallel graphene sheets, which both extend to $+/-$ infinity in the $z$ direction. In the simulations, to avoid edge effects, the graphene sheets are assigned periodic boundary conditions in the $x$ direction. The cross-plane current is injected from lead 1 and collected from leads 3 and 4. Figure 4b shows the conductance spectra of $C_{60}$, $C_{70}$, $C_{76}$, and $C_{90}$ as a function of the electrode Fermi level. In the experiments, the Fermi level is influenced by imperfections in the graphene and contact with copper. Several transmission spectra due to different orientations and locations were calculated and a Boltzmann distribution based on the total energy of the system was used to compute the average conductance (See Supplementary Fig. 30.). For a 0.1 eV

range of Fermi energies in the vicinity of the DFT-predicted Fermi level, we obtain the same trend as the experiment: $C_{60} < C_{70} < C_{76} < C_{90}$. This LUMO-dominated electron transport has been reported elsewhere for $C_{60}$[37,38]. The increase in conductance with the number of carbon atoms is plotted in Fig. 4c and is in a qualitative agreement with the experiment. The calculations further suggested that the $C_{70}$ junctions prefer the configuration with the longer axis parallel to the surface than the perpendicular configurations, since in the parallel configuration there is a stronger binding energy between $C_{70}$ and graphene surfaces. The stronger coupling between $C_{70}$ and graphene, as well as the greater density of $C_{70}$ molecular orbital energies near the HOMO and LUMO, further leads to higher transmission coefficient than $C_{60}$ within HOMO–LUMO gap (See Supplementary Discussion and Supplementary Figs. 32–34 for more details.).

To investigate the effect of conjugation and doping, the average conductance values (See Supplementary Fig. 31.) for molecules $C_{50}H_{10}$ and $C_{60}$ are shown in Fig. 4d, while for $C_{120}$ and $(C_{59}N)_2$ are in Fig. 3e. It is found that there is a large energy window between HOMO and LUMO for $C_{50}H_{10}$ where the conductance of $C_{60}$ is larger than that of $C_{50}H_{10}$ due to the significant difference in their HOMO–LUMO gaps. This theoretical result supports the experimental measurements, where $C_{60}$ is more conductive and shows that the broken conjugation of $C_{50}H_{10}$ due to the connection to ten hydrogen atoms leads to a decrease in conductance. To highlight the effect of doping, Fig. 4e shows the conductance of $(C_{59}N)_2$ vs. Fermi energy and reveals that the presence of nitrogen creates two transport resonances around $-0.5$ eV within the HOMO–LUMO gap of $C_{120}$. Consequently, over a large energy window, the conductance is significantly larger than that of the parent compound before nitrogen doping. We further analyze the corresponding frontier orbitals HOMO-2, HOMO-1, HOMO, and LUMO as plotted in Fig. 4f (More orbitals are shown in Supplementary Table 2.). Both the HOMO-1 and

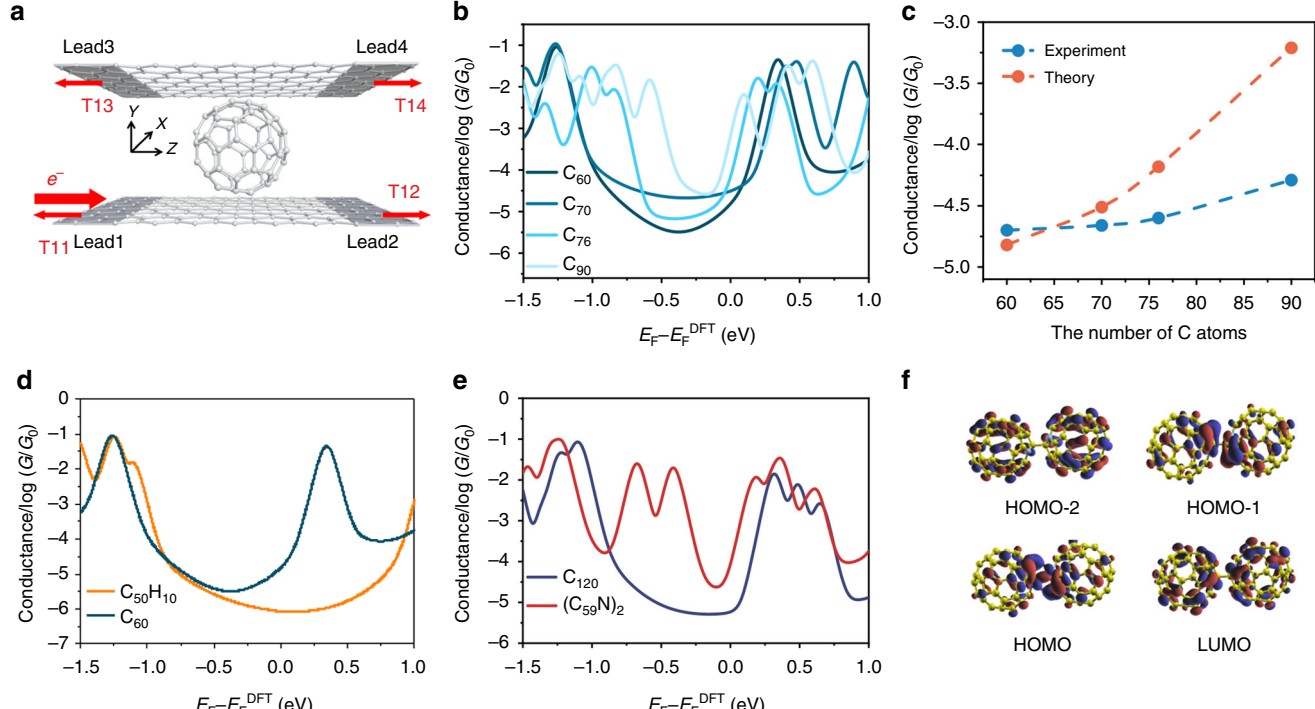

**Fig. 4** Density functional theory (DFT)-based charge transport investigations. **a** Schematic design of the vertical junction. **b** Conductance spectra as a function of Fermi level. The four curves stand for the average conductance of $C_{60}$, $C_{70}$, $C_{76}$, and $C_{90}$ over several different configurations, weighted by a Boltzmann distribution. **c** Conductance variance with the increasing size of fullerene molecules. The four red dots depict the conductance evolution for the four molecules shown in **b** when the Fermi level is aligned with that given by DFT. **d** Conductance spectra as a function of Fermi level. The two curves stand for the average conductance of $C_{50}H_{10}$ and $C_{60}$ over several different configurations weighted by a Boltzmann factor. **e** Conductance spectra as a function of Fermi level. The two curves present the average conductance of $C_{120}$ and $(C_{59}N)_2$ over several different configurations weighted by a Boltzmann distribution. Highest occupied molecular orbital (HOMO)-2-, HOMO-1-, HOMO-, and lowest unoccupied molecular orbital (LUMO)-mediated resonances are specified for $(C_{59}N)_2$. **f** Frontier molecular orbitals of isolate molecule $(C_{59}N)_2$: HOMO-2, HOMO-1, HOMO, LUMO, which corresponds to the specified resonances shown in **e**

HOMO are mainly localized on nitrogen, suggesting that the localized states on the two N atoms dominate the two peaks around −0.5 eV.

## Discussion

To conclude, we construct atomically defined 0D–2D hybrid single fullerene molecular junctions using copper-supported graphene electrodes and investigated their charge transport using a new type cross-plane MCBJ technique and DFT. This work demonstrates that HOMO–LUMO gap engineering of fullerenes allows the fine tuning of electron transmission through hybrid 0D–2D fullerene–graphene molecular devices, which are composed of only carbon atoms by >250%. The conductance of the graphene/single-fullerene/graphene junctions can be tuned more than one order of magnitude via inserting variety of full-erenes. We believe that the control of angstrom-scale charge transport, through the fabrication of atomically precise single-fullerene junctions on copper supported graphene electrodes, constitutes a critical step towards future all-carbon, dangling-bond-free electronics.

## Methods

**The preparation of graphene electrode and MCBJ measurement**. We have developed a new method to prepare a "graphene chip," in which a copper wire (0.3 mm in diameter) with graphene on the surface (6 Carbon Technology, Shenzhen, China) was bent into a curved surface, then fixed on a flexible substrate (10 mm × 30 mm, thickness 0.2 mm) with epoxy glue. Parallel to it, in the opposite side, another electrode treated in the same way was fixed on the substrate. With the help of a microscope, the bent part of the two copper wires are placed in the middle of

the substrate by the "head-to-head" configuration, and they are adjusted close enough (10–20 μm) but without touching (The photo and schematic of graphene chip are shown in Supplementary Fig. 1a, b.). Raman spectroscopy was employed to characterize the graphene electrodes (Supplementary Fig. 4) that the 2D peak near 2700/cm is sharp and symmetrical with a single Lorentz peak. The intensity of 2D peak is higher than that of G peak near 1600/cm, suggesting that the graphene on the copper wire we use was monolayer according to the previous report[39].

As in our recent works[40,41], Supplementary Fig. 2c, d show the MCBJ set-up and the schematic of the formation of the graphene/fullerene/graphene junction. The as-prepared sample chips are installed on the MCBJ set-up, and a polytef liquid cell is placed above the sample chips. The middle of the chip and the liquid cell is fixed by screws and steel sheets and is connected to the piezo below. At the beginning of the measurement, a stepping motor together with the piezo stack pulls down the middle part of the chip. As shown in Supplementary Fig. 1b, the two electrodes on the chip would gradually approach each other until the current increases to the set high trigger. Then the piezo stack starts to separate the two contacted graphene electrodes until the current falls below the detection limit of the MCBJ set-up. The displacement–conductance curves are repeated for thousands of cycles, and the conductance–displacement traces are recorded for further statistical analysis. The measurement is in a solution of fullerene in decane (98%, Aladdin, Shanghai, China). To prevent the effect of the conductance signal produced by the binding of graphene due to its π–π stacking interaction, the high conductance trigger was set to $3.2 \times 10^{-3}$ $G_0$.

In order to calibrate the stretching rate of graphene electrodes and the length of conductance plateau, we designed a graphene-electrode break junction configuration using vertical STM-based break junction (STM-BJ) protocol to measure the tunneling decay constant in the pure solvent (decane). As shown in Supplementary Fig. 2a, a bent copper wire and sheet (Both of them are coated with CVD graphene.) were used as tip and substrate, respectively, for STM-BJ processes. The absolute distance of the piezo, which was used to drive the tip, was calculated through the recorded data of driving voltages. We used this "graphene STM-BJ" protocol to calibrate the stretching distance in pure decane without target molecules and calculated the relative stretching displacement of graphene electrode from $1.0 \times 10^{-4}$ to $1.0 \times 10^{-6}$ $G_0$ to be ≈0.27 nm (Supplementary Fig. 2b). This indicates that the tunneling decay constant between graphene electrodes under

pure solvent condition is $\log[\Delta G/G_0]/\Delta z = -2/0.27$ nm $= -7.4$/nm. Therefore, before each graphene–molecule–graphene MCBJ test, the pure solvent should be done as a control experiment, thus the conductance signal was ensured to come from the target molecule. At last, using this obtained tunneling decay constant, we were able to calibrate the stretching distance of the horizontal graphene electrodes in MCBJ test. A detailed test process and data processing method can be found in refs. [40,41].

We also add the control experiments using "graphene STM-BJ" protocol (Supplementary Fig. 2a) in pure solvents (decane) without molecules. The graphene electrodes were "hard-contacted" with the high set trigger at 10 $G_0$. The conductance–displacement curves from both opening and closing processes were recorded, and several individual traces are shown in Supplementary Fig. 3. Because the graphene electrodes on the tip would be deformed after hard-contact, each curve was measured using a new tip.

All the experiments were performed at room temperature (under air conditioning set at $25 \pm 3$ °C).

**Theoretical methods**. Geometrical optimizations were performed by using the standard Kohn–Sham self-consistent density functional code SIESTA[35], with a generalized gradient approximation (GGA-PBE) exchange-correlation functional, and a double-$\zeta$ polarized atomic-orbital basis set for carbon. The cutoff energy was 200 Ry, and the force tolerance was 0.02 eV/Å. To compute their electrical conductance, the fullerenes were each placed between two graphene electrodes. For each structure, the transmission coefficient $T(E)$ describing the propagation of electrons of energy $E$ from lead 1 to lead $j$ ($j = 2, 3, 4$) was calculated using Gollum the quantum transport code[36], which utilizes the DFT mean-field Hamiltonian and overlap matrices from SIESTA and computes $T(E)$ via the following formula:

$$T(E) = Tr\left[\Gamma_1(E)g(E)\Gamma_j(E)g^\dagger(E)\right] \quad (1)$$

where $g(E)$ is the retarded Green's function of the molecule in the presence of the electrodes and $\Gamma_j(E) = i(\Sigma_j(E) - \Sigma_j^\dagger(E))/2$ is the anti-Hermitian part of the self-energies $\Sigma_j(E)$. $\Gamma_j$ determines the broadening of transmission resonances due to the contact between the molecule and electrode $j$. The finite-temperature conductance is obtained from the transmission coefficient via the following formula:

$$G = G_0 \int_{-\infty}^{+\infty} dE \, T(E)\left(-\frac{\partial f(E)}{\partial E}\right) \quad (2)$$

where $G_0 = 2e^2/h$ is the conductance quantum; $h$ is the Planck's constant; $e$ is the charge of an electron; $f(E) = (1 + \exp(E - E_F/k_BT))^{-1}$ is the Fermi-Dirac probability distribution function, $E_F$ is the Fermi energy, $T$ is the temperature, and $k_B$ is Boltzmann's constant. If the total energy of a molecule $i$ with a certain location and orientation on the electrodes is $E_i$, then the probability of obtaining such a molecular location and orientation is $P_i = A^{-1}e^{-E_i/k_BT}$, where $A = \sum_i e^{-E_i/k_BT}$ is the partition function. As is well known from statistical mechanics, if $G_i$ is the corresponding conductance, then the Boltzmann average of the conductance is $G = \sum_i P_i G_i$.

For each molecule, the transmission coefficient was computed for a range of contact geometries between graphene and the fullerene, examples of which are shown in Supplementary Fig. 29. Finally, the average conductance, weighted by the Boltzmann distribution based on total energies of each junction configuration, is obtained from formulae (2–6) in Supplementary Note 3.

In the experiments, the copper can modify the density of states in the graphene, primarily by shifting the Fermi energy of the graphene relative to the Dirac point. To account for this, we plot conductance values (Fig. 4) as a function of the Fermi energy. If a copper layer is connected to an external electrode and the current passes from the copper to the molecule via the graphene, then there will be an extra resistance due to the copper–graphene interface. However, since the area of the copper–graphene interface is much larger than the footprint of the molecule, this resistance will be negligible compared with the molecular resistance. Therefore, our calculation captures the dominant contribution to the junction resistance.

Also, the four-terminal simulation is used, because if the leads are terminated close to the fullerene, then multiple scattering between the lead edges and the fullerene will occur, which creates additional structure in the transmission curves vs. energy. In the experiment, there is no such termination and therefore the four-terminal calculation is closer to the experiment.

**Combustion synthesis of $C_{50}H_{10}$, $C_{60}$, $C_{70}$, $C_{76}$, and $C_{90}$**. The fullerene-containing soot was synthesized by means of an acetylene–benzene–oxygen diffusion combustion in our homemade set-up[42]. The flame was maintained under a pressure of 10–20 torr in the homemade set-up with the burner that consisted of two concentric tubes with the same center. Oxygen was fed to the inner tube, while benzene flowed through the space between the two concentric tubes. The synthetic conditions were optimized to give an optimal yield of typical $C_{60}$ in the soot and to keep the diffusion flame stable if possible. The optimized gas flow rates are 0.55 L/min for $O_2$, 1.10 L/min for $C_2H_2$, and 1.0–1.1 L/min for vaporous benzene. Both $C_{60}$ and $C_{70}$ are the major products with minor components of $C_{50}H_{10}$, $C_{76}$, and $C_{90}$, subject to high-performance liquid chromatography (HPLC) separation.

**Synthesis of $C_{120}$**. A "vibrating mill" was used for dimerization of $C_{60}$ in the absence of any solvent, based on previous literature[43,44]. This method was originally designed for the preparation of a well-mixed homogeneous powder by vigorously vibrating a stainless-steel capsule that contains the sample and stainless-steel balls with a frequency of 2800 cycles/min. A mixture of $C_{60}$ and 20 molar equivalents of KCN powder was vigorously vibrated for 30 min under nitrogen. The major product is dimer of $C_{60}$ with unchanged $C_{60}$. After separation using HPLC on a 5PBB column, the purified $C_{120}$ was afforded (Supplementary Fig. 13). The purified sample was confirmed via mass spectrometry (Supplementary Fig. 20) and UV-Vis absorption spectrometry (Supplementary Fig. 27).

**Synthesis of $(C_{59}N)_2$**. Heterofullerene $C_{59}N$ was synthesized and stabilized as dimer $(C_{59}N)_2$ according to previous literature[45–49]. We optimized the synthesis conditions to increase the yield of $(C_{59}N)_2$. As shown in Supplementary Fig. 14, HPLC was performed on a buckyprep column with toluene as the eluent to separate the product. The purified sample was confirmed via mass spectrometry (Supplementary Fig. 21) and UV-Vis absorption spectrometry (Supplementary Fig. 28).

**HPLC separation and purification of $C_{50}H_{10}$, $C_{60}$, $C_{70}$, $C_{76}$, $C_{90}$, $C_{120}$, and $(C_{59}N)_2$**. The fullerene-containing carbon soot was extracted with toluene by ultrasound and followed by filtration. The compositions of the toluene-extracting solutions were very complex and typically contain lots of polycyclic aromatic hydrocarbons, fullerenes, and fullerene derivatives. Several stages of HPLC with different columns were conducted to separate and purify the sample of $C_{50}H_{10}$, $C_{60}$, $C_{70}$, $C_{76}$, and $C_{90}$ from the toluene-extracting solutions of the crude soot. In each stage, the $C_{50}H_{10}$-, $C_{60}$-, $C_{70}$-, $C_{76}$-, and $C_{90}$-containing fractions were collected and followed by concentration and HPLC until the purified sample with 99.9% purity were obtained. Supplementary Figs. 8–12 show the separation columns, elution conditions, and retention time of $C_{50}H_{10}$, $C_{60}$, $C_{70}$, $C_{76}$, and $C_{90}$ for the multi-stage HPLC isolation. All the separations were performed at room temperature using toluene as the eluent. After several cycles of separation, the purified samples were obtained. The hydrofullerene ($C_{50}H_{10}$) and pristine fullerenes ($C_{60}$, $C_{70}$, and $C_{76}$) can be identified by mass spectrometry (Supplementary Figs. 15–18) and UV-Vis absorption spectrometry (Supplementary Figs. 22–15), as only one isomer was isolated for each of them. It was worth mentioning that four isomers of $C_{90}$ were isolated from the crude soot. The higher fullerene $C_{90}$ isomer that we used for research herein have the same mass spectrum (Supplementary Fig. 19) and UV-Vis absorption spectrum (Supplementary Fig. 2) to the results previously published for their isomer $C_I(30)$-$C_{90}$[50] [Note that the isomeric $C_{90}$ can be specified by the Fowler–Manolopoulos nomenclature[51], but we omit the symmetry and spiral code for clarification in the main text.]. Our X-ray crystallographic experiment further identified the precise structure of $C_I(30)$-$C_{90}$ involved.

**Mass spectra of $C_{50}H_{10}$, $C_{60}$, $C_{70}$, $C_{76}$, $C_{90}$, $C_{120}$, and $(C_{59}N)_2$**. Mass spectra were recorded by a Bruker HCT ion trap instrument interfaced by an atmospheric pressure chemical ionization (APCI) source. In the APCI, ion–molecule reactions were occurred to generate the ions at ambient conditions, and then the ions were transferred into the mass analyzer by a vacuum interface. An oven with temperature tunable in the range of 200–350 °C was connected to the APCI source for solvent evaporation and possibly sample decomposition.

## Data availability

The data that support the findings of this study are available from the corresponding author upon reasonable request.

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

## Acknowledgements

This work was supported by the National Key R&D Program of China (2017YFA0204902), National Natural Science Foundation of China (Nos. 21722305, 21673195, 21703188), and Young Thousand Talent Project of China. S.Y.X. is particularly grateful to the financial supports from NSFC (21721001 and 51572231) and the Major Science and Technology Project between University-Industry Cooperation in Fujian Province (2016H6023). Support from the UK EPSRC is acknowledged through grant Nos. EP/M014452/1, EP/P027156/1, EP/N017188/1, and EP/N03337X/1 and from European Union's Horizon 2020 Research and Innovation Program under grant agreement No. 767187, 'QuIET'. H.S. acknowledges the UKRI Future Leaders Fellowship No. MR/S015329/1 and the Leverhulme Trust Early Career Fellowship No. ECF-2017-186. H.R.T., Y.Z.T. and S.Y.X. thank Prof. Shangfeng Yang at USTC for his generous help in synthesis of C120.

## Author contributions

W.H., S.Y.X., C.L., and Z.X. designed the experiments and co-supervised the project. W.H., C.L., Z.Tan, and Q.W. wrote the manuscript with inputs from all authors. D.Z., Z.Tan, and J.P. carried out the break junction experiments and analyzed the data. H.T., S.Y.X., and Y.Z.T., were responsible for molecular synthesis and characterization. W.H., Z.Tang, Z.B.C., and J.S. built the electrical measurement instrument and wrote the software to control the break junction set-up. Y.Y., J.P., and Z.X. prepared the graphene microchip and optimized the break junction set-up. Q.W., C.L., H.S., and S.H. performed the theoretical modeling. J.L. participated in the revision of the manuscript. All authors conceived the work and discussed the experiments.

## Additional information

**Competing interests:** The authors declare no competing interests.

