## [Peer Review File · Nature Communications]

Reviewers' comments:

Reviewer #1 (Remarks to the Author):

In this manuscript, Tan et al. investigated charge transport in various graphene/single-fullerene/graphene junctions. Authors successfully synthesized C60, C70, C76, C90, and C50H10, C120, and (C59N)₂. All purified fullerene family is identified by careful characterizations with UV-Vis Absorption spectrometry and mass spectrometry. Authors used CVD-graphene coated on copper wires as electrodes and performed experiments on all carbon junctions at the single-molecule level with MCBJ technique. They observed that conductance increases as the size of fullerene and attributed the trend to the decreased band gap between HOMO and LUMO. In addition, they investigated interrupted conjugation with C50H10 and showed that the conductance in interrupted conjugation is significantly lower than C60. Lastly, they compared the conductance between C120 and (C59N)₂ and demonstrated that substituting C with N noticeably increases conductance due to the presence of two transport resonances. In addition, all experimental observations are corroborated by DFT based calculation.

Thorough investigation with well-established MCBJ technique and computation is impressive and clearly presented. Finding from this work is convincing and insights obtained from this work are of interest in molecular electronics and beyond. I feel that this manuscript deserves to be published in Nature Communications after a revision. Please see below my comments to improve understanding of the manuscript.

1. While experimental procedure is well described, explanation of DFT simulation is somewhat lacking. The reviewer suggests elaborating DFT simulations in method section. Standard DFT is known to underestimate the HOMO-LUMO gap and overestimate conductance. Specify whether you used standard DFT or an extension version of DFT such as DFT+ Σ .
2. Authors wrote that transmission is obtained from different orientations and locations. Authors need to write more specifically what they mean by different orientation and locations.
3. To obtain average conductance, average weighted by Boltzmann has been used. It needs the explanation as to why this type of analysis is used.
4. I suggest to change the all conductance expression as the form of 2×10^{-5} G Ω not $10^{-4.7}$ G Ω .
5. Authors mentioned that high trigger is set as $10^{-2.5}$ G Ω to avoid graphene electrode damage. What is the point-contact conductance of graphene-graphene? Experiment require graphene-graphene contact needs to be made first, since the step in conductance traces is observed.

Minor comments

6. Figure 3e caption refers to the inset, but inset is missing.
7. Line 160: what do you mean by "latter"?
8. Line 188: If theory is referred, it needs to be red dots, not blue dots. Check this again.
9. For STM-BJ, did authors use copper sheet coated with CVD graphene? Specify the condition to avoid any confusions.
- 10 For experiments, is it performed under ambient condition? Or inert condition?

Reviewer #2 (Remarks to the Author):

This paper demonstrated nicely full-carbon electronics at angstrom scale by building various graphene/single-fullerene/graphene junctions and studying the charge transport properties experimentally and theoretically. A relatively systematic study was carried out by changing the

inserted fullerene structures. The main result of this work is that: 1 The conductance of single pristine fullerene increase with the size of fullerenes ($C_{60} < C_{70} < C_{76} < C_{90}$), which is in contrast with normal length dependence observed for most conjugated organic molecules; 2 disrupting the conjugation ($C_{50}H_{10}$) significantly reduces charge transport efficiency; 3 heteroatoms doping ($(C_{59}N)_2$) can introduce extra resonances between the original HOMO and LUMO, enhancing the single molecule conductance. Overall, the paper describes interesting observations that will be critical for future study of full-carbon electronics, the major conclusions are well supported by the data, and the manuscript is well written. I recommend publication of the manuscript after minor revision regarding the following points.

1. The range of conductance studied in this paper starts from $10E-2.5$ G0. But in previous conductance studies of single C_{60} sandwiched between metal electrodes (e.g. Kiguchi. et.al, J. Phys. Chem. C, Vol. 112, No. 22, 2008), the observed conductance is much higher even than the high trigger set in the present work. Have the authors tried looking at a higher conductance range to make sure that there is no other conductance state out of the conductance range studied?
2. In the study of pristine fullerenes, the increased conductance was merely attributed to the decreased bandgap with molecular size, which is not quite accurate. In most organic conjugated systems, the bandgap also decreases with the increasing size of the conjugation system, however, the conductance decreases with molecular length, which is in contrast to the observation in the present study. Another important reason should be that for pristine fullerene structures, the reduce in bandgap is much sharper than the growth in molecular size, which makes the major difference between a globular conjugating system with a linearly conjugated system. An illustration on the competing relation between the bandgap and molecular size or a simple conductance estimation based on the two trends can be helpful for the readers to understand the key reason.
3. If relative displacement represents the size of the inserted molecule, then not until the end of the plateau (~ 1 nm), should the molecule be fully inserted between the two electrodes (Fig. 1a). For example, at the beginning of each plateau (~ 0 nm), the molecule should be only started to migrate between the two electrodes and only a small part of the fullerene is sandwiched and involved in the charge transport, since the distance of the gap is not enough. In this case, the junction configuration should be substantially different than Fig. 1a. However, the conductance measured from 0 nm to 1 nm is indeed not changing substantially, suggesting the same configuration for the whole process. What's the authors' interpretation?
4. It is kind of expected that C_{76} and C_{70} show two conductance state or wider conductance distribution since they are more ellipsoid structures with a short and a long axis. But the experiment only observed one state with similar/narrower distribution than C_{60} . It would be good if the authors can extend a comment or discussion on this.
5. In contrast with the prediction in the beginning of the paper, the theoretical study realized that the bandgap of C_{70} is larger than that of C_{60} (Fig. 4b). The actual reason of the higher conductance of C_{70} than C_{60} as illustrated by the theoretical study should be a crossover between the transmission function of C_{60} and C_{70} (probably due to coupling?). A discussion should be provided for the bandgap and the transmission crossover to rationale the experimental observations.

Reviewer #3 (Remarks to the Author):

Tan et al. report in "Atomically defined angstrom-scale all-carbon junctions" a systematic investigation of fullerenes ($C_{50}H_{10}$, C_{60} , C_{70} , C_{76} , C_{90} , C_{120} , $(C_{59}N)_2$) in graphene/Cu break junctions. The result on C_{60} (conductance of $10^{-4.7}$ G0) should be put in relation to existing measurements (e.g. CPL (2009) 477, 189, Guo et al., where a conductance of about 10^{-2} G0 was found, or Nano Lett., 2008, 8, 1291, Neel et al., with contact conductances of 10^{-1} G0 in UHV). It turns out that the conductance increases for large fullerenes, which is attributed to the smaller HOMO-LUMO gap in

larger fullerenes. Undoubtedly this is important information (I like C₅₀H₁₀ which is, unfortunately a fairly good insulator), though if the paper should be published in Nature Communications, it should discuss on what it signifies for "all carbon electronics". It has to be stressed that the experiments, though nice, do not deal with "all-carbon electronics" since the electrodes in the break junctions are graphene on copper.

Minor comments:

I like the junction picture in Figure Supplementals S1(d) more than Fig. 1 (a).

What says theory (and experiment) on whether the conductance is electron or hole like?

It should be said somewhere at which temperature the experiment has been performed.

Maybe it is useful to determine the effective conductance in considering the intrinsic junction resistance without the molecules, also I would like to read on how big the conductance of a pure Cu-Cu junction is?

There are as well other one dimensional Carbon nanoribbons that are discussed in view of "all-carbon electronics"

Reviewer #4 (Remarks to the Author):

In this manuscript the authors study single-molecule junctions of different fullerenes between graphene electrodes. In contrast with previously reported results with graphene electrodes, the current flows perpendicular to the graphene. In this novel and interesting configuration, they investigate the influence on charge transport of gap size by probing pristine fullerenes of increasing number of atoms, and of the breaking of the conjugation and of doping. The paper is well-written and presents relevant results in the field of molecular electronics opening a new line of research with graphene electrodes.

I recommend publication after the following points are properly addressed.

1. The experimental setup is incompletely described, the author should give more details of the experimental setup, in particular thickness of copper wires. I am also missing a characterization of the CVD graphene electrodes, are they monolayer graphene? Bilayer?
2. No results for the conductance of the closed junction, that is graphene in contact with graphene, are given, as the junction is never fully closed. I understand that this could damage the graphene layers but it is worth exploring.
3. The theoretical model used in the simulations does not properly describe the experimental configuration. In the experiment the electrodes consist of graphene on copper, while in the simulations simple graphene electrodes are considered. This is important because the copper will certainly modify the density of states of the graphene. The use of these simple graphene electrodes makes also necessary to consider an unrealistic 4-terminal arrangement in the simulations. It is essential to use for the simulation graphene on copper electrodes and a cross-plane current in a 2-terminal configuration. These deficiencies of the model could be responsible for the poor agreement of the theoretical data to the experimental data. For example, as shown in Fig. 4c, the conductance increases by up to half an order of magnitude with the increasing size of the fullerenes, whereas the experimental conductance increases by almost 2 orders of magnitude.
4. Additionally in line 213 and following the copper wire with the CVD graphene layer is termed "chip" which I find quite inadequate.

Point-to-point reply to reviewers' comments

COMMENTS TO AUTHOR:

Reviewer 1:

General comments: In this manuscript, Tan et al. investigated charge transport in various graphene/single-fullerene/graphene junctions. Authors successfully synthesized C_{60} , C_{70} , C_{76} , C_{90} , and $C_{50}H_{10}$, C_{120} , and $(C_{59}N)_2$. All purified fullerene family is identified by careful characterizations with UV-Vis Absorption spectrometry and mass spectrometry. Authors used CVD-graphene coated on copper wires as electrodes and performed experiments on all carbon junctions at the single-molecule level with MCBJ technique. They observed that conductance increases as the size of fullerene and attributed the trend to the decreased band gap between HOMO and LUMO. In addition, they investigated interrupted conjugation with $C_{50}H_{10}$ and showed that the conductance in interrupted conjugation is significantly lower than C_{60} . Lastly, they compared the conductance between C_{120} and $(C_{59}N)_2$ and demonstrated that substituting C with N noticeably increases conductance due to the presence of two transport resonances. In addition, all experimental observations are corroborated by DFT based calculation.

Thorough investigation with well-established MCBJ technique and computation is impressive and clearly presented. Finding from this work is convincing and insights obtained from this work are of interest in molecular electronics and beyond. I feel that this manuscript deserves to be published in *Nature Communications* after a revision. Please see below my comments to improve understanding of the manuscript.

Comment 1. While experimental procedure is well described, explanation of DFT simulation is somewhat lacking. The reviewer suggests elaborating DFT simulations in Methods section. Standard DFT is known to underestimate the HOMO-LUMO gap and

overestimate conductance. Specify whether you used standard DFT or an extension version of DFT such as DFT+ Σ .

Reply:

We thank the reviewer for this suggestion and have added the following detailed description of the DFT simulations to the Methods section.

“Theoretical methods

Geometrical optimizations were performed by using the standard Kohn-Sham self-consistent density functional code SIESTA³⁵, with a generalized gradient approximation (GGA-PBE) exchange-correlation functional, and a double- ζ polarized atomic-orbital basis set for carbon. The cutoff energy was 200 Ry, and the force tolerance was 0.02 eV/Å. To compute their electrical conductance, the fullerenes were each placed between two graphene electrodes. For each structure, the transmission coefficient $T(E)$ describing the propagation of electrons of energy E from lead 1 to lead j ($j=2, 3, 4$) was calculated using Gollum the quantum transport code³⁶, which utilises the DFT mean-field Hamiltonian and overlap matrices from SIESTA and computes $T(E)$ via the following formula:

$$T(E) = \text{Tr}[\Gamma_1(E)g(E)\Gamma_j(E)g^\dagger(E)] \quad (1)$$

where $g(E)$ is the retarded Green’s function of the molecule in the presence of the electrodes and $\Gamma_j(E) = i(\Sigma_j(E) - \Sigma_j^\dagger(E))/2$ is the anti-Hermitian part of the self-energies $\Sigma_j(E)$. Γ_j determines the broadening of transmission resonances due to the contact between the molecule and electrode j . The finite-temperature conductance is obtained from the transmission coefficient via the following formula:

$$G = G_0 \int_{-\infty}^{+\infty} dE T(E) \left(-\frac{\partial f(E)}{\partial E}\right) \quad (2)$$

where $G_0 = 2e^2/h$ is the conductance quantum; h is the Planck’s constant; e is the charge of an electron; $f(E) = (1 + \exp(E - E_F/k_B T))^{-1}$ is the Fermi-Dirac

probability distribution function, E_F is the Fermi energy, T is the temperature and k_B is Boltzmann's constant.”

We agree that standard DFT could underestimate the HOMO-LUMO gap and overestimate the conductance. However since our aim is to predict trends rather than actual conductance values, standard DFT was used. This predicts an increasing conductance with the fullerene size, in agreement with the experimental results. It also shows that transport is LUMO dominated as reported in the literature (*Phys. Rev. Lett.* **2010**, 104, 036103.).

Comment 2. Authors wrote that transmission is obtained from different orientations and locations. Authors need to write more specifically what they mean by different orientation and locations.

Reply:

To be more precise, we added Fig. S29 to the Supplementary Information, showing all the configurations of the various orientations and locations of the fullerenes on the electrodes.

C_{70}

C_{76}

C_{90}

Fig. S29 | Schematic of all the configurations of the various orientations and locations of the fullerenes on the electrodes.

And we also added the following sentences to the section of Theoretical methods:

“For each molecule, the transmission coefficient was computed for a range of contact geometries between graphene and the fullerene, examples of which are shown in Fig. S29 of the Supplementary Information. Finally, the average conductance, weighted by the Boltzmann distribution based on total energies of each junction configuration is obtained from formulae (2-6) in the SI (section VII).”

Comment 3. To obtain average conductance, average weighted by Boltzmann has been used. It needs the explanation as to why this type of analysis is used.

Reply:

To clarify this point, the following text has been added to the Methods section of the manuscript.

“If the total energy of a molecule i with a certain location and orientation on the electrodes is E_i , then the probability of obtaining such a molecular location and orientation is $P_i = Z^{-1} e^{-E_i/k_B T}$, where $Z = \sum_i e^{-E_i/k_B T}$ is the partition function. As well-known from statistical mechanics, if G_i is the corresponding conductance, then the Boltzmann average of the conductance is $G = \sum_i P_i G_i$.”

Comment 4. I suggest to change the all conductance expression as the form of $2 \times 10^{-5} G_0$ not $10^{-4.7} G_0$.

Reply:

Thank you for this suggestion. We have revised the expression of conductance value in the following places of the paper.

In the main text we now write:

“..., the typical individual conductance-displacement curves of C_{60} show clear plateaus around $2.0 \times 10^{-5} G_0$...”

“..., which indicate the most probable conductance peak of graphene/single-C₆₀, C₇₀, C₇₆, and C₉₀/graphene located at $2.0 \times 10^{-5} G_0$, $2.2 \times 10^{-5} G_0$, $2.5 \times 10^{-5} G_0$ and $5.1 \times 10^{-5} G_0$ respectively...”

“The conductance of the low-conjugated graphene/single-C₅₀H₁₀/graphene junctions located at $2.2 \times 10^{-6} G_0$, ...”

“Fig. 3e shows that the most probable conductance of graphene/single-C₁₂₀/graphene junctions is $1.3 \times 10^{-5} G_0$...”

“...the conductance of graphene/single-(C₅₉N)₂/graphene (Fig. 3d for the 2D conductance histogram and conductance plateau distribution) increases by 150% and reaches $2.1 \times 10^{-5} G_0$ (Fig. 3e),...”

In the caption of Fig. 1

“...the high trigger was set as $3.2 \times 10^{-3} G_0$...The top right insert is the relative displacement distribution ranging from 3.2×10^{-4} to $4.0 \times 10^{-6} G_0$...”

In the caption of Fig. 2

“(The conductance range to determine the displacement are from 3.2×10^{-4} to $6.3 \times 10^{-7} G_0$)”

In the caption of Fig. 3

“(The conductance range to determine the displacement of C₁₂₀ and (C₅₉N)₂ are from 3.2×10^{-4} to $4.0 \times 10^{-6} G_0$, $1.0 \times 10^{-5} G_0$ respectively)”

And in the Methods section we write:

“To prevent the effect of the conductance signal produced by the binding of graphene due to its own π - π stacking interaction, the high conductance trigger was set as $3.2 \times 10^{-3} G_0$.”

“...and calculated the relative stretching displacement of graphene electrode from 1.0×10^{-4} to $1.0 \times 10^{-6} G_0$ to be ≈ 0.27 nm...”

And in the Supplementary Information Fig. S2 caption

“...the top right insert is the relative displacement distribution from 1.0×10^{-4} to $1.0 \times 10^{-6} G_0$.”

In the caption of Fig. S3

“...the top right insert is the relative displacement distribution ranging from 1×10^{-4} to $1 \times 10^{-6} G_0$.”

In the caption of Fig. S4

“The top right insert is the relative displacement distribution, from a to c, the statistics range from 3.2×10^{-4} to $1.0 \times 10^{-5} G_0$, $6.3 \times 10^{-6} G_0$, $1.6 \times 10^{-5} G_0$ respectively.”

Comment 5. Authors mentioned that high trigger is set as $10^{-2.5} G_0$ to avoid graphene electrode damage. What is the point-contact conductance of graphene-graphene? Experiment require graphene-graphene contact needs to be made first, since the step in conductance traces is observed.

Reply:

As also mentioned by reviewer 4, the hard contact between two graphene electrodes will lead to the deformation of the graphene electrodes and damage of the graphene layers. When the higher trigger was applied, we found that the MCBJ process cannot continue and the opening and closing precesses could rarely be found. Thus the MCBJ process based on the graphene chip developed in this work should be carried out in the “soft-contact” mode, which is similar to the $I(s)$ and $I(t)$ of STM measurement (*J. Am. Chem. Soc.* **2003**, 125, 15294. *Nat. Mater.* **2006**, 5, 995. *J. Am. Chem. Soc.* **2015**, 137, 9971. *Nat. Commun.* **2011**, 2, 305.).

To further clarify this point, we have added the control experiments using “graphene STM-BJ” protocol (Supplementary Information Fig. S2a) in pure solvents (decane) without molecules. The graphene electrodes were “hard-contact” with the conductance of around $10 G_0$. The conductance-displacement curves from both opening and closing process were recorded and several individual traces are shown in new Fig. S3. Because the graphene electrodes on the tip would be deformed after hard contact, each curve was measured using a new tip. Most of the closing and opening curves show a similar conductance feature between $10^{-2} G_0$ and $10^{-3} G_0$ (marked in the dark red square), which corresponds to the state in which the two graphene electrodes are contacting each other. Meanwhile, the opening traces show the relatively long distance to have a hard contact. It should also be noticed that the displacement to break the graphene-graphene contact

is tens of nanometers, which is significantly larger than the π - π interaction distance (usually considered to be about 0.35 nanometers), suggesting the deformation of the graphene layers from the copper surface may happen at the hard-contact mode. Below $10^{-3} G_0$, the direct tunneling curves demonstrate that the electrodes are separated and the current decreases exponentially with the increase of distance. Due to the random contact angle and the deformation of the two graphene electrodes, the point contact conductances of graphene vary a lot from sample to sample, but the higher set trigger of $3.2 \times 10^{-2} G_0$ is below the determined conductance range of the contracted graphene electrodes. The data also show that the range from $3.2 \times 10^{-2} G_0$ to $1.00 \times 10^{-6} G_0$ provides a wide enough window to measure the conductance of graphene/fullerene/graphene junctions.

We added the above results to the Methods section:

“We also add the control experiments using “graphene STM-BJ” protocol (Supplementary Information Fig. S2a) in pure solvents (decane) without molecules. The graphene electrodes were “hard-contacted” with the high set trigger at $10 G_0$. The conductance-displacement curves from both opening and closing processes were recorded, and several individual traces are shown in Supplementary Information Fig. S3. Because the graphene electrodes on the tip would be deformed after hard-contact, each curve was measured using a new tip.”

And added a new figure in the Supplementary Information:

Fig. S3 | The individual conductance-displacement curves of a control experiment of graphene electrodes with the high set trigger at $10 G_0$ (The left panel is the closing process and the right panel is the opening process).

And relevant discussion in Supplementary Information:

“As shown in Fig. S3, most of the closing and opening curves show a similar conductance feature between $10^{-2} G_0$ and $10^{-3} G_0$ (marked in the dark red square), which corresponds to the state in which the two graphene electrodes are contacting each other. Meanwhile, the opening traces show the relatively long distance to achieve hard contact. It should also be noticed that the displacement to break the graphene-graphene contact is tens of nanometers, which is significantly larger than the π - π interaction distance (usually considered to be about 0.35 nanometers), suggesting the deformation of the graphene layers from the copper surface may happen with hard-contact mode. After $10^{-3} G_0$, the direct tunneling curves demonstrate that the electrodes are separated and the current decreases exponentially with the increase of distance. Due to the random contact angle and the deformation of the two graphene electrodes, the point contact conductance of graphene vary a lot from sample to sample, but the set higher trigger of $3.2 \times 10^{-2} G_0$ is below the determined conductance range of the contracted graphene electrodes. The data also show that the range from $3.2 \times 10^{-2} G_0$ to $1.0 \times 10^{-6} G_0$ provides a wide enough window to measure the conductance of graphene/fullerene/graphene junctions.”

Minor comments

Comment 6. Figure 3e caption refers to the inset, but inset is missing.

Reply:

We are sorry for our mistake, and add the missing insert in Fig. 3e.

The revised Fig. 3 is as follows:

Comment 7. Line 160: what do you mean by “latter”?

Reply:

This refers to ‘the electrode Fermi level’ mentioned at the ends of the previous sentence. To clarify this point, the sentence has been modified by replacing ‘latter’ with ‘Fermi level’.

“Fig. 4b shows the conductance spectra of C_{60} , C_{70} , C_{76} and C_{90} as a function of the electrode Fermi level. In the experiments, the Fermi level is influenced by imperfections in the graphene and contact with copper.”

Comment 8. Line 188: If theory is referred, it needs to be red dots, not blue dots. Check this again.

Reply:

Thank reviewer for pointing this error, ‘blue’ has been corrected to ‘red’.

Comment 9. For STM-BJ, did authors use copper sheet coated with CVD graphene? Specify the condition to avoid any confusions.

Reply:

Yes, we use a copper sheet coated with CVD graphene for STM-BJ. To express this more clearly, we revised the sentence in the Methods section.

“As shown in Supplementary Information Fig. S2a, a bent copper wire and sheet (Both of them are coated with CVD graphene) were used as tip and substrate respectively for STM-BJ processes.”

Comment 10. For experiments, is it performed under ambient condition? Or inert condition?

Reply:

All the the experiments were done in air and room temperature, and we have added the sentence in the Methods section.

“All the the experiments were performed in air and at room temperature (under air conditioning set at 25 ± 3 °C).”

Reviewer 2:

General comments: This paper demonstrated nicely full-carbon electronics at angstrom scale by building various graphene/single-fullerene/graphene junctions and studying the charge transport properties experimentally and theoretically. A relatively systematic study was carried out by changing the inserted fullerene structures. The main result of this work is that: The conductance of single pristine fullerene increase with the size of fullerenes ($C_{60} < C_{70} < C_{76} < C_{90}$), which is in contrast with normal length dependence observed for most conjugated organic molecules; 2. Disrupting the conjugation ($C_{50}H_{10}$) significantly reduces charge transport efficiency; 3. Heteroatoms doping ($(C_{59}N)_2$) can introduce extra resonances between the original HOMO and LUMO, enhancing the single molecule conductance. Overall, the paper describes interesting observations that will be critical for future study of full-carbon electronics, the major conclusions are well supported by the data, and the manuscript is well written. I recommend publication of the manuscript after minor revision regarding the following points.

Comment 1. The range of conductance studied in this paper starts from $10E^{-2.5} G_0$. But in previous conductance studies of single C_{60} sandwiched between metal electrodes (e.g. Kiguchi. et.al, *J. Phys. Chem. C*, Vol. 112, No. 22, **2008**), the observed conductance is

much higher even than the high trigger set in the present work. Have the authors tried looking at a higher conductance range to make sure that there is no other conductance state out of the conductance range studied?

Reply:

We thank the reviewer for the kind suggestions. We tried to look into the high conductance range by using a higher trigger of the conductance. However, the higher conductance requires a hard contact between two graphene electrodes, leading to the deformation of the graphene electrodes and damage of the graphene layers. When the higher trigger was applied, we found that the MCBJ process cannot continue, and useful opening and closing curves could rarely be found.

Thus the MCBJ process based on the graphene chip developed in this work should be carried out in the “soft-contact” mode, which is similar to the $I(s)$ and $I(t)$ of STM measurement (*J. Am. Chem. Soc.*, **2003**, 125, 15294. *Nat. Mater.*, **2006**, 5, 995. *J. Am. Chem. Soc.*, **2015**, 137, 9971. *Nat. Commun.*, **2011**, 2, 305.).

To further clarify this point, we have added the control experiments using “graphene STM-BJ” protocol (Supplementary Information Fig. S2a) in pure solvents (decane) without molecules. The graphene electrodes were “hard-contact” with the conductance of around $10 G_0$. The conductance-displacement curves from both opening and closing process were recorded and several individual traces are shown in new Fig. S3. Because the graphene electrodes on the tip would be deformed after hard contact, each curve was measured using a new tip. Most of the closing and opening curves show a similar conductance feature between $10^{-2} G_0$ and $10^{-3} G_0$ (marked in the dark red square), which corresponds to the state in which the two graphene electrodes are contacting each other. Meanwhile, the opening traces show the relatively long distance to have a hard contact. It should also be noticed that the displacement to break the graphene-graphene contact is tens of nanometers, which is significantly larger than the π - π interaction distance (usually considered to be about 0.35 nanometers), suggesting the deformation of the graphene layers from the copper surface may happen at the hard-contact mode. Below $10^{-3} G_0$, the direct tunneling curves demonstrate that the electrodes are separated and

the current decreases exponentially with the increase of distance. Due to the random contact angle and the deformation of the two graphene electrodes, the point contact conductances of graphene vary a lot from sample to sample, but the higher set trigger of $3.2 \times 10^{-2} G_0$ is below the determined conductance range of the contracted graphene electrodes. The data also show that the range from $3.2 \times 10^{-2} G_0$ to $1.00 \times 10^{-6} G_0$ provides a wide enough window to measure the conductance of graphene/fullerene/graphene junctions.

We added the above results to the Methods section:

“We also add the control experiments using “graphene STM-BJ” protocol (Supplementary Information Fig. S2a) in pure solvents (decane) without molecules. The graphene electrodes were “hard-contacted” with the high set trigger at $10 G_0$. The conductance-displacement curves from both opening and closing processes were recorded, and several individual traces are shown in Supplementary Information Fig. S3. Because the graphene electrodes on the tip would be deformed after hard-contact, each curve was measured using a new tip.”

And added a new figure in the Supplementary Information:

Fig. S3 | The individual conductance-displacement curves of a control experiment of graphene electrodes with the high set trigger at $10 G_0$ (The left panel is the closing process and the right panel is the opening process).

And relevant discussion in Supplementary Information:

“As shown in Fig. S3, most of the closing and opening curves show a similar conductance feature between $10^{-2} G_0$ and $10^{-3} G_0$ (marked in the dark red square), which corresponds to the state in which the two graphene electrodes are contacting each other. Meanwhile, the opening traces show the relatively long distance to achieve hard contact. It should also be noticed that the displacement to break the graphene-graphene contact is tens of nanometers, which is significantly larger than the π - π interaction distance (usually considered to be about 0.35 nanometers), suggesting the deformation of the graphene layers from the copper surface may happen with hard-contact mode. After $10^{-3} G_0$, the direct tunneling curves demonstrate that the electrodes are separated and the current decreases exponentially with the increase of distance. Due to the random contact angle and the deformation of the two graphene electrodes, the point contact conductance of graphene vary a lot from sample to sample, but the set higher trigger of $3.2 \times 10^{-2} G_0$ is below the determined conductance range of the contracted graphene electrodes. The data also show that the range from $3.2 \times 10^{-2} G_0$ to $1.0 \times 10^{-6} G_0$ provides a wide enough window to measure the conductance of graphene/fullerene/graphene junctions.”

By comparing the conductance measurement in the pure solvent and with fullerenes, we can conclude that the newly measured conductance features varying from different fullerenes are from the graphene-fullerene-graphene junctions, which also coincide well with the displacement analysis and the DFT calculated transmissions. The lower conductance compared with metal/ C_{60} /metal junctions may be due to the presence of two interfaces via π - π interaction.

We further added some discussion in the main text:

“To enable a quantitative comparison, we construct the 1D conductance histograms shown in Fig. 1d, which indicate that the most probable conductance peaks of graphene/single- C_{60} , C_{70} , C_{76} , and C_{90} /graphene located at $2.0 \times 10^{-5} G_0$, $2.2 \times 10^{-5} G_0$, $2.5 \times 10^{-5} G_0$ and $5.1 \times 10^{-5} G_0$ respectively, demonstrating that there is a more than 250% conductance enhancement from C_{60} to C_{90} . This result shows that the conductance of the single- C_{60} junctions with graphene electrodes is lower than that of

gold electrodes²³⁻²⁵, which agrees with the previous report that the π - π interaction is enough to construct molecular junctions, but the conductance are reduced²⁶....”

And added some references:

23. Kiguchi, M. & Murakoshi, K. Conductance of single C₆₀ molecule bridging metal electrodes. *J. Phys. Chem. C* **112**, 8140-8143 (2008).

24. Guo, Y., Kiguchi, M., Zhao, J. W. & Murakoshi, K. Fabrication and conductance characterization of single C₆₀ molecular junction in solutions. *Chem. Phys. Lett.* **477**, 189-193 (2009).

25. Neel, N., Kroger, J., Limot, L. & Berndt, R. Conductance of oriented C₆₀ molecules. *Nano Lett.* **8**, 1291-1295 (2008)

26. Wu, S. M. *et al.* Molecular junctions based on aromatic coupling. *Nat. Nanotech.* **3**, 569-574 (2008).

Comment 2. In the study of pristine fullerenes, the increased conductance was merely attributed to the decreased bandgap with molecular size, which is not quite accurate. In most organic conjugated systems, the bandgap also decreases with the increasing size of the conjugation system, however, the conductance decreases with molecular length, which is in contrast to the observation in the present study. Another important reason should be that for pristine fullerene structures, the reduce in bandgap is much sharper than the growth in molecular size, which makes the major difference between a globular conjugating system with a linearly conjugated system. An illustration on the competing relation between the bandgap and molecular size or a simple conductance estimation based on the two trends can be helpful for the readers to understand the key reason.

Reply:

Thank you for your constructive suggestions. To accommodate this point, we have amended the text at the beginning of page 6 as follows:

“Further investigation suggests that the increase is correlated with the rapid decrease of the fullerene HOMO (Highest Occupied Molecular Orbital)-LUMO (Lowest Unoccupied Molecular Orbital) gap (The HOMO-LUMO gaps of fullerenes were

analyzed from the UV-Vis spectra, See Supplementary Information section V Table S1 in detail), which overcomes the tendency for electrical conductance to decrease with molecular length²⁸⁻³⁰. This suggests that the conductance of graphene/single-fullerene/graphene junctions could be fine-tuned by making further choices from the huge family of pristine fullerenes with structure-dependent bandgaps. An illustration of the competing relation between the bandgap and molecular size is provided in recent studies of the length-dependent conductance of porphyrin wires formed from oligoyne-linked porphyrin units or from fused porphyrin tapes^{31,32}. In both cases, their HOMO-LUMO gaps shrink with increasing length, but for oligoyne-linked wires, this decrease is rather slow and therefore their conductance decreases with length. In contrast, the HOMO-LUMO gap of the more strongly coupled porphyrin tapes decreases more rapidly with length and their conductance are found to increase with length.”

And add new References:

31. Leary, E. *et al.* Bias-driven conductance increase with length in porphyrin tapes. *J. Am. Chem. Soc.* **140**, 12877-12883 (2018).

32. Limburg, B. *et al.* Anchor groups for graphene-porphyrin single-molecule transistors. *Adv. Funct. Mater.* **28** (2018).

And the sentence as follows was revised.

“As shown in Fig. 1e, the conductance of the graphene/single-fullerene/graphene junctions increases with the size of fullerenes from C₆₀ with ~7 angstroms to C₉₀ with ~9 angstroms, which is opposite to the conductance decay with length in organic molecules based single-molecule junctions²²⁻²⁴.”

Comment 3. If relative displacement represents the size of the inserted molecule, then not until the end of the plateau (~ 1 nm), should the molecule be fully inserted between the two electrodes (Fig. 1a). For example, at the beginning of each plateau (~ 0 nm), the molecule should be only started to migrate between the two electrodes and only a small part of the fullerene is sandwiched and involved in the charge transport, since the distance of the gap is not enough. In this case, the junction configuration should be substantially different than Fig. 1a. However, the conductance measured from 0 nm to

1 nm is indeed not changing substantially, suggesting the same configuration for the whole process. What's the authors' interpretation?

Reply:

Similar to the BJ process we described in our previous review (*Chem. Soc. Rev.*, 2015, **44**, 889-901), the process of forming the Au/molecule/Au junction is shown in the following schematic:

Fig. R1 | Scenario of the evolution of a molecular junction during BJ measurement.

Molecules are bridged between two gold electrodes via anchoring groups. With the increased separation of two pairs of electrodes, the molecule junctions went through various configurations and conductance information was monitored during the whole process. We believe that the formation of graphene/fullerene/graphene junctions also undergoes a similar process, and the following schematic diagram is provided to better understand the process during the separating of two graphene electrodes.

Fig. R2 | The schematic of the formation of graphene/fullerene/graphene junction.

1, 2 and 3 present the state in which the electrodes are about to be opened, part of the electrodes are opened, and the electrodes are fully opened, respectively. Fullerenes were assembled on graphene by π - π interaction before the electrodes were opened. In

this way, fullerenes provide an orbit for charge transport between electrodes at the moment the electrodes are opened, and the conductance of the whole molecular junction has been detected, and then the molecular junction gradually goes through 2, 3 configurations. Due to the decrease of through-space tunneling contribution and the reduced fullerene-graphene coupling, the conductance corresponding to the 2, 3 configurations shown in the figure may vary slightly with the increased distance between the two graphene electrodes.

We added the following schematic of the formation of graphene/fullerene/graphene junction in Supplementary Information Fig. S1:

Fig. S1 | **a, b**, The photo and schematic of graphene chip. **c**, The photo of our homemade MCBJ experiment setup. **d**, The schematic of the formation of graphene/fullerene/graphene junction.

We revised the sentence in the Methods section:

“As in our recent works^{40,41}, Supplementary Information Figs. S1c and S1d show the MCBJ setup and the schematic of the formation of graphene/fullerene/graphene junction...”

And we added the following discussion in Supplementary Information:

“Fig. S1d gives a schematic of the evolution of a graphene/fullerene/graphene junction. Panels 1, 2 and 3 present the state in which the electrodes are about to be opened, part

of the electrodes are opened, and the electrodes are fully opened, respectively. Fullerenes were assembled on graphene by π - π interaction before the electrodes were opened. In this way, fullerenes provide a pathway for charge transport between electrodes at the moment the electrodes are opened, and the conductance of the whole molecular junction has been detected, and then the molecular junction gradually goes through configurations 2, 3. Due to the decrease of through-space tunneling and the reduced fullerene-graphene coupling, the conductance corresponding to the configurations 2, 3 shown in the figure may vary slightly with the increasing distance between the two graphene electrodes.”

Comment 4. It is kind of expected that C_{76} and C_{70} show two conductance state or wider conductance distribution since they are more ellipsoid structures with a short and a long axis. But the experiment only observed one state with similar/narrower distribution than C_{60} . It would be good if the authors can extend a comment or discussion on this.

Reply:

Thank you for the suggestion. From the experimental viewpoint, the relative displacement distribution of C_{76} and C_{70} are slightly larger than that of C_{60} , suggesting that there is a certain probability that C_{70} can be transformed from the short axis perpendicular to graphene plane to the long axis during the stretching process. However, the probability that the long axis will be perpendicular to the graphene plane is much lower than that of the short axis perpendicular to the plane, because the binding energy of the former is lower. Consequently the less-probable long axis configuration has little effect on the conductance distribution.

To clarify this point, we carried out some further theoretical calculations. The results have been added to the Supplementary Information.

“Configurations corresponding to the longer axis oriented perpendicular to the surface (see Fig. S33) have a lower conductance and lower binding energy than configurations with the longer axis parallel to the surface.

Fig. S33 | A configuration of C_{70} with the long axis perpendicular to the surface

Fig. S34a below shows examples of conductance versus Fermi energy for different orientations of the molecule on the surface, each with the long axis parallel to the surface. Fig. S34b shows the same curves, but with additional examples obtained when the long axis is oriented perpendicular to the surface. When E_F is close to the DFT-predicted Fermi energy E_F^{DFT} (ie when $E_F - E_F^{DFT} = 0$) the spread of conductance values is barely increased by the inclusion of the less-probable additional examples. Furthermore, since their lower binding energy renders them less probable than configurations with the long axis parallel to the surface, they barely affect the Boltzmann-weighted average conductance, shown by the blue curves in Fig. S34a and b.

Fig. S34 | Conductance curves of different configurations of C_{70} . a, Conductance curves for 6 configurations in which the long axis is parallel to the surface. The blue line shows the Boltzmann average of these 6 curves. **b,** The conductance curves of fig. a, but with 4 more curves added, corresponding to configurations in which the long axis

is perpendicular to the surface. The blue line shows the Boltzmann average of these 10 curves.”

And we also added some sentence in the main text:

“..... The increase in conductance with the number of carbon atoms is plotted in Fig. 4c and is in a qualitative agreement with the experiment. The calculations further suggested that the C_{70} junctions prefer the configuration with the longer axis parallel to the surface than the perpendicular configurations, since in the parallel configuration there is a stronger binding energy between C_{70} and graphene surfaces. The stronger coupling between C_{70} and graphene, as well as the greater density of C_{70} molecular orbital energies near the HOMO and LUMO, further leads to higher transmission coefficient than C_{60} within HOMO-LUMO gap (See more details in Figs. S32-34 of Supplementary Information).”

Comment 5. In contrast with the prediction in the beginning of the paper, the theoretical study realized that the bandgap of C_{70} is larger than that of C_{60} (Fig. 4b). The actual reason of the higher conductance of C_{70} than C_{60} as illustrated by the theoretical study should be a crossover between the transmission function of C_{60} and C_{70} (probably due to coupling?). A discussion should be provided for the bandgap and the transmission crossover to rationale the experimental observations.

Reply:

DFT does not accurately predict energy gaps, because these are excited state properties and furthermore Kohn-Sham energy gaps are not the physical UV-Vis gaps. It is also reported that the DFT predicts that the Kohn-Sham HOMO-LUMO gap of C_{60} is slightly smaller than that of C_{70} (*Phys. Rev. Lett.* **1992**, 69, 69; S. M. Lee, *Chem. Phys. Lett.* **2005**, 404, 206.). On the other hand, our experimental results show that the UV-Vis gap of C_{70} is slightly smaller than that of C_{60} . Slight discrepancies of this kind are not uncommon.

The higher conductance of C_{70} is partly due to the higher coupling of the C_{70} to graphene and due to the greater density of C_{70} molecular orbital energies near the HOMO and LUMO. The greater coupling for C_{70} is illustrated by the plots below for one particular

orientation of the closest fullerene hexagon relative to the graphene. This shows, for example, that the HOMO-mediated resonance of C_{70} is wider than that of C_{60} , as indicated by the red and blue arrows. This difference originates from the stronger coupling between graphene and the C_{70} molecule. Furthermore, as shown by the density of states of the gas phase C_{60} and C_{70} in new Fig. S32d, there are more molecular orbital energies below the HOMO (-2 eV \sim -1 eV) and more states located near the LUMO ($0 \sim 0.5$ eV) for C_{70} (lower panel in new Fig. S32d) compared to C_{60} (upper panel in new Fig. S31d). These extra eigenstates contribute to the transmission coefficient even within the HOMO-LUMO gap and lead to an increase in the conductance of C_{70} , compared to C_{60} .

To clarify this point, the above text and figures have been added to the Supplementary Information.

“

Fig. S32 | The comparison of the transport properties between C_{60} and C_{70} . a, b, The configurations of C_{60} and C_{70} junction with similar contact geometry. Lateral view (upper panel) and top view (lower panel). **c,** The corresponding transmission functions. **d,** The density of states of the two gas phase molecules.

The higher conductance of C_{70} is partly due to the higher coupling of the C_{70} to graphene and due to the greater density of C_{70} molecular orbital energies near the HOMO and LUMO. The greater coupling for C_{70} is illustrated by the plots below for one particular orientation of the closest fullerene hexagon relative to the graphene. This shows, for example, that the HOMO-mediated resonance of C_{70} is wider than that of C_{60} , as indicated by the red and blue arrows. This difference originates from the stronger coupling between graphene and the C_{70} molecule. Furthermore, as shown by the density of states of the gas phase C_{60} and C_{70} in Fig. S32d, there are more molecular orbital energies below the HOMO (-2 eV \sim -1 eV) and more states located near the LUMO ($0 \sim 0.5$ eV) for C_{70} (lower panel in d) compared to C_{60} (upper panel in d). These extra eigenstates contribute to the transmission coefficient even within the HOMO-LUMO gap and lead to an increase in the conductance of C_{70} , compared to C_{60} .”

And we also added the following sentences in the main text:

“..... The increase in conductance with the number of carbon atoms is plotted in Fig. 4c and is in a qualitative agreement with the experiment. The calculations further suggested that the C_{70} junctions prefer the configuration with the longer axis parallel to the surface than the perpendicular configurations, since in the parallel configuration there is a stronger binding energy between C_{70} and graphene surfaces. The stronger coupling between C_{70} and graphene, as well as the greater density of C_{70} molecular orbital energies near the HOMO and LUMO, further leads to higher transmission coefficient than C_{60} within HOMO-LUMO gap (See more details in Figs. S32-34 of Supplementary Information).”

Reviewer 3:

General comments: Tan et al. report in "Atomically defined angstrom-scale all-carbon junctions" a systematic investigation of fullerenes ($C_{50}H_{10}$, C_{60} , C_{70} , C_{76} , C_{90} , C_{120} , $(C_{59}N)_2$) in graphene/Cu break junctions. The result on C_{60} (conductance of $10^{-4.7} G_0$) should be put in relation to existing measurements (e.g. CPL (2009) 477, 189, Guo et al., where a conductance of about $10^{-2} G_0$ was found, or Nano Lett., 2008, 8, 1291, Neel

et al., with contact conductances of $10^{-1} G_0$ in UHV). It turns out that the conductance increases for large fullerenes, which is attributed to the smaller HOMO-LUMO gap in larger fullerenes. Undoubtedly this is important information (I like C₅₀H₁₀ which is, unfortunately a fairly good insulator), though if the paper should be published in *Nature Communications*, it should discuss on what it signifies for "all carbon electronics". It has to be stressed that the experiments, though nice, do not deal with "all-carbon electronics" since the electrodes in the break junctions are graphene on copper.

Reply:

We agree with the reviewer that the conductance of fullerene molecular junctions measured by graphene electrodes is lower than that measured by metal electrodes. Refer to the reply of **comment 5 of reviewer 1**, the results confirm that the conductance of graphene/fullerene/graphene junctions is lower than that of point contact between two graphene sheets, and we believe that the lower conductance is also due to the existence of two π - π interaction interfaces in the graphene/fullerene/graphene junctions. Previous studies have demonstrated that π - π interaction is enough to construct molecular junctions but reduces their conductance (*Nat. Nanotech.*, **2008**, 3, 569.).

To address this point, we added the following discussion to the main text:

“To enable a quantitative comparison, we construct the 1D conductance histograms shown in Fig. 1d, which indicate the most probable conductance peak of graphene/single-C₆₀, C₇₀, C₇₆, and C₉₀/graphene are located at $2.0 \times 10^{-5} G_0$, $2.2 \times 10^{-5} G_0$, $2.5 \times 10^{-5} G_0$ and $5.1 \times 10^{-5} G_0$ respectively, demonstrating that there is a more than 250% conductance enhancement from C₆₀ to C₉₀. The result shows that the conductance of the single-C₆₀ junctions with graphene electrodes is lower than that of gold electrodes²³⁻²⁵, which agrees with the previous report that the π - π interaction is enough to construct molecular junctions, but the conductance is reduced²⁶...”

And added some references:

23. Kiguchi, M. & Murakoshi, K. Conductance of single C₆₀ molecule bridging metal electrodes. *J. Phys. Chem. C* **112**, 8140-8143 (2008).

24. Guo, Y., Kiguchi, M., Zhao, J. W. & Murakoshi, K. Fabrication and conductance characterization of single C₆₀ molecular junction in solutions. *Chem. Phys. Lett.* **477**, 189-193 (2009).

25. Neel, N., Kroger, J., Limot, L. & Berndt, R. Conductance of oriented C₆₀ molecules. *Nano Lett.* **8**, 1291-1295 (2008)

26. Wu, S. M. *et al.* Molecular junctions based on aromatic coupling. *Nat. Nanotech.* **3**, 569-574 (2008).

Raman spectroscopy was employed to characterize graphene electrodes and to determine the number of graphene layers. The data show that the 2D peak near 2700 cm⁻¹ is sharp and symmetrical, and has a perfect single Lorentz peak. And its intensity is higher than G peak near 1600 cm⁻¹. According to the previous report (*Nano Res.*, 2008, **1**, 273), these indicate that the graphene on the copper wire we use was a monolayer. We added the Raman spectroscopy data to Supplementary Information.

Fig. S4 | Raman characterization of CVD graphene on copper wire.

And some sentence in the Methods section:

“...With the help of a microscope, the bent part of the two copper wires are placed in the middle of the substrate by the “head-to-head” configuration, and they are adjusted close enough (10-20 μm) but without touch (The photo and schematic of graphene chip are shown in Supplementary Information Fig. S1a, b). Raman spectroscopy was employed to characterize the graphene electrodes (Fig. S4 in Supplementary

Information) that the 2D peak near 2700 cm^{-1} is sharp and symmetrical with a single Lorentz peak. The intensity of 2D peak is higher than that of G peak near 1600 cm^{-1} , suggesting that the graphene on the copper wire we use was monolayer according to the previous report³⁹.”

Relevant reference was added:

39. Ni, Z. H., Wang, Y. Y., Yu, T. & Shen, Z. X. Raman spectroscopy and imaging of graphene. *Nano Res.* **1**, 273-291 (2008).

Concerning the questions on the term of “all-carbon electronics”, we agree that the copper can modify the density of states in the graphene, primarily by shifting the Fermi energy of the graphene relative to the Dirac point. To account for this, we plot conductance (Fig. 4) as a function of the Fermi energy. If a copper layer is connected to an external electrode and the current passes from the copper to the molecule via the graphene, then there will be an extra resistance due to the copper-graphene interface. However, since the area of the copper-graphene interface is much larger than the footprint of the molecule, this resistance will be negligible compared with the molecular resistance. Therefore, our calculation captures the dominant contribution to the junction resistance. Moreover, according to previous studies of molecular junctions with different metal electrodes (*J. Am. Chem. Soc.* **2010**, 132, 756. *Nano Lett.* **2013**, 13, 3358.), the effect of different metal substrates on the charge transport through the junction should be relatively small.

To accommodate these points and avoid the confusion, we have added more detailed theoretical methods.

“In the experiments, the copper can modify the density of states in the graphene, primarily by shifting the Fermi energy of the graphene relative to the Dirac point. To account for this, we plot conductance (Fig. 4) as a function of the Fermi energy. If a copper layer is connected to an external electrode and the current passes from the copper to the molecule via the graphene, then there will be an extra resistance due to the copper-graphene interface. However, since the area of the copper-graphene interface is much larger than the footprint of the molecule, this resistance will be

negligible compared with the molecular resistance. Therefore our calculation captures the dominant contribution to the junction resistance.”

Concerning the significance of this work for “all carbon electronics”, the aim of this work is to achieve the controllability of graphene-fullerene-graphene devices through bandgap engineering of atomically defined carbon allotropes, which could be crucial towards the electronics mainly composed of carbon atoms.

To further clarify this point, we revised the conclusion:

“To conclude, we constructed atomically defined 0D-2D hybrid single fullerene molecular junctions using copper supported graphene electrodes, and investigated their charge transport using a new type cross-plane MCBJ technique and density functional theory.”

“We believe that the fabrication and charge transport control of the angstrom-scale and atomically precise single fullerene junctions on copper supported graphene electrodes constitute a critical step towards all-carbon electronics in the future benefiting from numerous dangling-bond-free fullerenes.”

Minor comments:

Comment 1. I like the junction picture in Figure Supplementals S1(d) more than Fig. 1 (a).

Reply:

Thank you for your suggestion. We have replaced the schematic of graphene/single-fullerene/graphene junction in Fig. 1a (main text) with a slightly modified Fig. S1d (Supplementary Information) for better demonstration of the scales of the graphene electrodes and fullerenes.

Comment 2. What says theory (and experiment) on whether the conductance is electron or hole like?

Reply:

We have discussed this issue in the DFT simulations section of the article, where we note:

“This LUMO-dominated electron transport has been reported elsewhere for C_{60} ^{37,38}.”

When C_{60} is connected to electrodes, the electron transfer from electrodes to C_{60} and occupies the LUMO due to its strong electronegativity, which leads to electron-transport (*Phys. Rev. B* **2001** 63, 121104). Literature (*Phys. Rev. B* **2014** 90, 245404) also shows C_{70} is LUMO-mediated electron transport. That is to say, LUMO-dominated transport indicates electron transport while HOMO-dominated transport corresponds to hole transport.

Comment 3. It should be said somewhere at which temperature the experiment has been performed.

Reply:

We thank the reviewer for the constructive comment. All the the experiments were done in air and room temperature, and we added the sentence in the Methods section.

“All the experiments were done in air and at room temperature (under air conditioning set at 25 ± 3 °C).”

Comment 4. Maybe it is useful to determine the effective conductance in considering the intrinsic junction resistance without the molecules, also I would like to read on how big the conductance of a pure Cu-Cu junction is?

Reply:

Thanks for these suggestions. To further clarify this point, we have added the control experiments using “graphene STM-BJ” protocol (Supplementary Information Fig. S2a) in pure solvents (decane) without molecules. The graphene electrodes were “hard-contact” with the conductance of around $10 G_0$. The conductance-displacement curves from both opening and closing process were recorded and several individual traces are shown in new Fig. S3. Because the graphene electrodes on the tip would be deformed after hard contact, each curve was measured using a new tip. Most of the closing and opening curves show a similar conductance feature between $10^{-2} G_0$ and $10^{-3} G_0$ (marked in the dark red square), which corresponds to the state in which the two graphene electrodes are contacting each other. Meanwhile, the opening traces show the relatively long distance to have a hard contact. It should also be noticed that the displacement to break the graphene-graphene contact is tens of nanometers, which is significantly larger than the π - π interaction distance (usually considered to be about 0.35 nanometers), suggesting the deformation of the graphene layers from the copper surface may happen at the hard-contact mode. Below $10^{-3} G_0$, the direct tunneling curves demonstrate that the electrodes are separated and the current decreases exponentially with the increase of distance. Due to the random contact angle and the deformation of the two graphene electrodes, the point contact conductances of graphene vary a lot from sample to sample, but the higher set trigger of $3.2 \times 10^{-2} G_0$ is below the determined conductance range of the contracted graphene electrodes. The data also show that the range from $3.2 \times 10^{-2} G_0$ to $1.00 \times 10^{-6} G_0$ provides a wide enough window to measure the conductance of graphene/fullerene/graphene junctions.

We added the above results to the Methods section:

“We also add the control experiments using “graphene STM-BJ” protocol (Supplementary Information Fig. S2a) in pure solvents (decane) without molecules.

The graphene electrodes were “hard-contacted” with the high set trigger at $10 G_0$. The conductance-displacement curves from both opening and closing processes were recorded, and several individual traces are shown in Supplementary Information Fig. S3. Because the graphene electrodes on the tip would be deformed after hard-contact, each curve was measured using a new tip.”

And added a new figure in the Supplementary Information:

Fig. S3 | The individual conductance-displacement curves of a control experiment of graphene electrodes with the high set trigger at $10 G_0$ (The left panel is the closing process and the right panel is the opening process).

And relevant discussion in Supplementary Information:

“As shown in Fig. S3, most of the closing and opening curves show a similar conductance feature between $10^{-2} G_0$ and $10^{-3} G_0$ (marked in the dark red square), which corresponds to the state in which the two graphene electrodes are contacting each other. Meanwhile, the opening traces show the relatively long distance to achieve hard contact. It should also be noticed that the displacement to break the graphene-graphene contact is tens of nanometers, which is significantly larger than the π - π interaction distance (usually considered to be about 0.35 nanometers), suggesting the deformation of the graphene layers from the copper surface may happen with hard-contact mode. After $10^{-3} G_0$, the direct tunneling curves demonstrate that the electrodes are separated and the current decreases exponentially with the increase of distance. Due to the random

contact angle and the deformation of the two graphene electrodes, the point contact conductance of graphene vary a lot from sample to sample, but the set higher trigger of $3.2 \times 10^{-2} G_0$ is below the determined conductance range of the contracted graphene electrodes. The data also show that the range from $3.2 \times 10^{-2} G_0$ to $1.0 \times 10^{-6} G_0$ provides a wide enough window to measure the conductance of graphene/fullerene/graphene junctions.”

For the conductance of a pure Cu-Cu junction, some previous work (*Nano res.* **2017**, 10, 3314. *Appl. Phys. A* **2005**, 81, 1539. *Nanotechnology* **2007**, 18, 424011. *J. Am. Chem. Soc.* **2008**, 130, 13228.) have demonstrated that the conductance of copper atom point contact is similar to that of Au at G_0 . We did not observe the conductance peak around G_0 in hard-contact mode, this indicates that copper is not pulled out because graphene is coated.

Comment 5. There are as well other one dimensional Carbon nanoribbons that are discussed in view of "all-carbon electronics".

Reply:

Thank you for your suggestion, and the newly published paper on Carbon nanoribbons are added to the introduction section as references:

“All-carbon electronics^{1,2}, mostly employing 1D carbon nanotubes (CNTs) as transport materials and 2D graphene as electrodes, can operate much faster and thus hold significant promise beyond silicon electronics^{3,4}. Although the CNTs and carbon nanoribbons⁵ have been demonstrated to fabricate computing circuits⁶, the ability of carbon allotropes to conduct charge is sensitive to atomic structural topology..... ”

5. Slota, M. *et al.* Magnetic edge states and coherent manipulation of graphene nanoribbons. *Nature* **557**, 691-696 (2018).

Reviewer 4:

General comments: In this manuscript the authors study single-molecule junctions of different fullerenes between graphene electrodes. In contrast with previously reported results with graphene electrodes, the current flows perpendicular to the graphene. In

this novel and interesting configuration, they investigate the influence on charge transport of gap size by probing pristine fullerenes of an increasing number of atoms, and the breaking of the conjugation and doping. The paper is well-written and presents relevant results in the field of molecular electronics opening a new line of research with graphene electrodes.

I recommend publication after the following points are properly addressed.

Comment 1. The experimental setup is incompletely described, the author should give more details of the experimental setup, in particular, thickness of copper wires. I am also missing a characterization of the CVD graphene electrodes, are they monolayer graphene? Bilayer?

Reply:

Thank you for this suggestion. Specific material parameters and detailed MCBJ process are given in the Methods section to address the questions, and we revised the Methods section about the preparation of graphene electrode and MCBJ measurement.

“We have developed a new method to prepare a “graphene chip,” in which a copper wire (0.3 mm in diameter) with graphene on the surface (6 Carbon Technology, Shenzhen, China) was bent into a curved surface, then fixed on a flexible substrate (10 mm × 30 mm, thickness 0.2 mm) with epoxy glue. Parallel to it, in the opposite side, another electrode treated in the same way was fixed on the substrate. With the help of a microscope, the bent part of the two copper wires are placed in the middle of the substrate by the “head-to-head” configuration, and they are adjusted close enough (10-20 μm) but without touch (The photo and schematic of graphene chip are shown in Supplementary Information Fig. S1a, b)...

As in our recent works^{40,41}, Supplementary Information Figs. S1c and S1d show the MCBJ setup and the schematic of the formation of the graphene/fullerene/graphene junction. The as-prepared sample chips are installed on the MCBJ set-up, and a polytetrafluoroethylene liquid cell is placed above the sample chips. The middle of the chip and the liquid cell is fixed by screws and steel sheets and is connected to the piezo below. At the beginning of the measurement, a stepping motor together with the piezo stack pulls down the

middle part of the chip. As shown in Fig. S1b, the two electrodes on the chip would gradually approach each other until the current increases to the set high trigger. Then the piezo stack starts to separate the two contacted graphene electrodes until the current falls below the detection limit of the MCBJ set-up. The displacement-conductance curves are repeated for thousands of cycles, and the conductance-displacement traces are recorded for further statistical analysis. The measurement is in a solution of fullerene in decane (98%, Aladdin, Shanghai, China). To prevent the effect of the conductance signal produced by the binding of graphene due to its π - π stacking interaction, the high conductance trigger was set to $3.2 \times 10^{-3} G_0$.”

Raman spectroscopy was employed to characterize graphene electrodes and to determine the number of graphene layers. The data show that the 2D peak near 2700 cm^{-1} is sharp and symmetrical, and has a perfect single Lorentz peak. And its intensity is higher than G peak near 1600 cm^{-1} . According to the previous report (*Nano Res.* 2008, **1**, 273), these indicate that the graphene on the copper wire we use was a monolayer. We added the Raman spectroscopy data to Supplementary Information.

Fig. S4 | Raman characterization of CVD graphene on copper wire.

And some sentence in the Methods section:

“...With the help of a microscope, the bent part of the two copper wires are placed in the middle of the substrate by the “head-to-head” configuration, and they are adjusted close enough (10-20 μm) but without touch (The photo and schematic of graphene chip are shown in Supplementary Information Fig. S1a, b). Raman spectroscopy was

employed to characterize the graphene electrodes (Fig. S4 in Supplementary Information) that the 2D peak near 2700 cm^{-1} is sharp and symmetrical with a single Lorentz peak. The intensity of 2D peak is higher than that of G peak near 1600 cm^{-1} , suggesting that the graphene on the copper wire we use was monolayer according to the previous report³⁹.”

Relevant reference was added:

39. Ni, Z. H., Wang, Y. Y., Yu, T. & Shen, Z. X. Raman spectroscopy and imaging of graphene. *Nano Res.* **1**, 273-291 (2008).

Comment 2. No results for the conductance of the closed junction, that is graphene in contact with graphene, are given, as the junction is never fully closed. I understand that this could damage the graphene layers but it is worth exploring.

Reply:

Thank you for this suggestion. As you mentioned, the hard contact between two graphene electrodes will lead to the deformation of the graphene electrodes and damage of the graphene layers. When the higher trigger was applied, we found that the MCBJ process cannot continue and there is rarely opening and closing curves could be found. Thus the MCBJ process based on graphene chip developed in this work should be carried out in the “soft-contact” mode, which is similar to the $I(s)$ and $I(t)$ of STM measurement (*J. Am. Chem. Soc.*, **2003**, 125, 15294. *Nat. Mater.*, **2006**, 5, 995. *J. Am. Chem. Soc.*, **2015**, 137, 9971. *Nat. Commun.*, **2011**, 2, 305.).

To further clarify this point, we have added the control experiments using “graphene STM-BJ” protocol (Supplementary Information Fig. S2a) in pure solvents (decane) without molecules. The graphene electrodes were “hard-contact” with the conductance of around $10 G_0$. The conductance-displacement curves from both opening and closing process were recorded and several individual traces are shown in new Fig. S3. Because the graphene electrodes on the tip would be deformed after hard contact, each curve was measured using a new tip. Most of the closing and opening curves show a similar conductance feature between $10^{-2} G_0$ and $10^{-3} G_0$ (marked in the dark red square), which corresponds to the state in which the two graphene electrodes are contacting each other.

Meanwhile, the opening traces show the relatively long distance to have a hard contact. It should also be noticed that the displacement to break the graphene-graphene contact is tens of nanometers, which is significantly larger than the π - π interaction distance (usually considered to be about 0.35 nanometers), suggesting the deformation of the graphene layers from the copper surface may happen at the hard-contact mode. Below $10^{-3} G_0$, the direct tunneling curves demonstrate that the electrodes are separated and the current decreases exponentially with the increase of distance. Due to the random contact angle and the deformation of the two graphene electrodes, the point contact conductances of graphene vary a lot from sample to sample, but the higher set trigger of $3.2 \times 10^{-2} G_0$ is below the determined conductance range of the contracted graphene electrodes. The data also show that the range from $3.2 \times 10^{-2} G_0$ to $1.00 \times 10^{-6} G_0$ provides a wide enough window to measure the conductance of graphene/fullerene/graphene junctions.

We added the above results to the Methods section:

“We also add the control experiments using “graphene STM-BJ” protocol (Supplementary Information Fig. S2a) in pure solvents (decane) without molecules. The graphene electrodes were “hard-contacted” with the high set trigger at $10 G_0$. The conductance-displacement curves from both opening and closing processes were recorded, and several individual traces are shown in Supplementary Information Fig. S3. Because the graphene electrodes on the tip would be deformed after hard-contact, each curve was measured using a new tip.”

And added a new figure in the Supplementary Information:

Fig. S3 | The individual conductance-displacement curves of a control experiment of graphene electrodes with the high set trigger at $10 G_0$ (The left panel is the closing process and the right panel is the opening process).

And relevant discussion in Supplementary Information:

“As shown in Fig. S3, most of the closing and opening curves show a similar conductance feature between $10^{-2} G_0$ and $10^{-3} G_0$ (marked in the dark red square), which corresponds to the state in which the two graphene electrodes are contacting each other. Meanwhile, the opening traces show the relatively long distance to achieve hard contact. It should also be noticed that the displacement to break the graphene-graphene contact is tens of nanometers, which is significantly larger than the π - π interaction distance (usually considered to be about 0.35 nanometers), suggesting the deformation of the graphene layers from the copper surface may happen with hard-contact mode. After $10^{-3} G_0$, the direct tunneling curves demonstrate that the electrodes are separated and the current decreases exponentially with the increase of distance. Due to the random contact angle and the deformation of the two graphene electrodes, the point contact conductance of graphene vary a lot from sample to sample, but the set higher trigger of $3.2 \times 10^{-2} G_0$ is below the determined conductance range of the contracted graphene electrodes. The data also show that the range from $3.2 \times 10^{-2} G_0$ to $1.0 \times 10^{-6} G_0$ provides a wide enough window to measure the conductance of graphene/fullerene/graphene junctions.”

We also analyzed the data of the closing processes of graphene/single-C₆₀/graphene junctions. The new figure was added in the Supplementary Information.

Fig. S7 | Data of the closing processes of graphene/single-C₆₀/graphene junctions. a, b, 2D and 1D conductance histogram of the closing processes of graphene/single-C₆₀/graphene junctions. c, The relative displacement distribution determined from 3.2×10^{-4} to $4.0 \times 10^{-6} G_0$.

As shown in new Fig. S7a. 2D conductance histogram of the closing processes of graphene/single-C₆₀/graphene junctions also shows a well-defined conductance feature. The relative displacement distributions of conductance plateaus (Fig. S7c) is similar to that of the opening process (Fig. 1c insert). Fig. S5b display that the most probable conductance peak of closing processes of graphene/single-C₆₀/graphene junctions located at $5.0 \times 10^{-5} G_0$, which is slightly larger than that of opening processes due to the contribution of pure tunneling to conductance (*Chem* **2019**, 5, 390.).

We added the following discussion to the main text:

“To enable a quantitative comparison, we construct the 1D conductance histogram shown in Fig. 1d, which indicate the most probable conductance peak of graphene/single-C₆₀, C₇₀, C₇₆, and C₉₀/graphene located at $2.0 \times 10^{-5} G_0$, $2.2 \times 10^{-5} G_0$, $2.5 \times 10^{-5} G_0$ and $5.1 \times 10^{-5} G_0$ respectively, demonstrating that there is a more than 250% conductance enhancement from C₆₀ to C₉₀. This result shows that the conductance of the single-C₆₀ junctions with graphene electrodes is lower than that of gold electrodes²³⁻²⁵, which agrees with the previous report that the π - π interaction is enough to construct molecular junctions, but the conductance are reduced²⁶. Meanwhile, the data of the closing process of graphene/single-C₆₀/graphene junctions are given in

Supplementary Information Fig. S7, which indicates that closing process form molecular junctions with the same length as the opening process, but the conductance ($5.0 \times 10^{-5} G_0$) is slightly larger than that of the opening process because of the contribution of pure tunneling²⁷...”

And added the reference:

27. Liu, J. Y. *et al.* Transition from tunneling leakage current to molecular tunneling in single-molecule junctions. *Chem* **5**, 390-401 (2019).

Comment 3. The theoretical model use in the simulations does not properly describe the experimental configuration. In the experiment the electrodes consist of graphene on copper, while in the simulations simple graphene electrodes are considered. This is important because the copper will certainly modify the density of states of the graphene. The use of these simple graphene electrodes makes also necessary to consider an unrealistic 4-terminal arrangement in the simulations. It is essential to use for the simulation graphene on copper electrodes and a cross-plane current in a 2-terminal configuration. This deficiencies of the model could be responsible for the poor agreement of the theoretical data to the experimental data. For example, as shown in Fig. 4c, the conductance increases by up to half an order of magnitude with the increasing size of the fullerenes, whereas the experimental conductance increases by almost 2 orders of magnitude.

Reply:

We thank the reviewer for this suggestion. Yes, we agree that the copper can modify the density of states in the graphene, primarily by shifting the Fermi energy of the graphene relative to the Dirac point. To account for this, we have plotted conductances (Fig. 4) as a function of the Fermi energy. If a copper layer is connected to an external electrode and the current passes from the copper to the molecule via the graphene, then there will be an extra resistance due to the copper-graphene interface. However, since the area of the copper-graphene interface is much larger than the footprint of the molecule, this resistance will be negligible compared with the molecular resistance. Therefore our calculation captures the dominant contribution to the junction resistance.

Also, the 4-terminal simulation is realistic, because if the leads are terminated close to the fullerene, then multiple scattering between the lead edges and the fullerene will occur, which creates additional structure in the transmission curves versus energy. In the experiment, there is no such termination, and therefore the four-terminal calculation is closer to the experiment. The theoretical conductance increases more quickly than the experimental data. This is almost certainly due to the ideal configurations modeled theoretically, which ignores unknown defects and impurities on the surfaces of the graphene electrodes.

To accommodate these points and avoid the confusion, we have added more detailed theoretical methods.

“In the experiments, the copper can modify the density of states in the graphene, primarily by shifting the Fermi energy of the graphene relative to the Dirac point. To account for this, we plot conductance values (Fig. 4) as a function of the Fermi energy. If a copper layer is connected to an external electrode and the current passes from the copper to the molecule via the graphene, then there will be an extra resistance due to the copper-graphene interface. However, since the area of the copper-graphene interface is much larger than the footprint of the molecule, this resistance will be negligible compared with the molecular resistance. Therefore our calculation captures the dominant contribution to the junction resistance.

Also, the 4-terminal simulation is used, because if the leads are terminated close to the fullerene, then multiple scattering between the lead edges and the fullerene will occur, which creates additional structure in the transmission curves versus energy. In the experiment, there is no such termination and therefore the four-terminal calculation is closer to the experiment.”

Comment 4. Additionally in line 213 and following the copper wire with the CVD graphene layer is termed “chip” which I find quite inadequate.

Reply:

Thank you for this suggestion. To express more clearly, we have revised the sentences describing the preparation of graphene electrodes in the Methods section.

“We have developed a new method to prepare “graphene chip” and the details are described as follows. A copper wire (0.3 mm in diameter) with graphene on the surface (6 Carbon Technology, Shenzhen, China) was bent into a curved surface, then fixed on a flexible substrate (10 mm × 30 mm, thickness 0.2 mm) with epoxy glue....”

REVIEWERS' COMMENTS:

Reviewer #1 (Remarks to the Author):

Authors have significantly improved the manuscript. Thus, I recommend publication in Nature Communications.

Reviewer #2 (Remarks to the Author):

All my concerns are addressed in the revised manuscript. In my opinion, the manuscript can be published.

Reviewer #3 (Remarks to the Author):

The authors answered my points and I can agree on publication, though they might like to correct for the flaw in the axis labeling of new Figure S3. Likely the axis units are $\log(G/G_0)$?

Reviewer #4 (Remarks to the Author):

The authors have responded satisfactorily to all the questions raised in my previous report. I am happy to recommend publication of the manuscript in its present form. I would like to note a typo in figure S3: the vertical axis should be "Conductance/ $\log(G/G_0)$ ".

Point-by-point response to reviewers' comments

Reviewer 3:

Comments: The authors answered my points and I can agree on publication, though they might like to correct for the flaw in the axis labeling of new Figure S3. Likely the axis units are $\log(G/G_0)$?

Reply:

Thank reviewer for pointing this error. We are sorry for our mistake, and revise the Supplementary Figure 3 as follow:

Supplementary Figure 3 | The individual conductance-displacement curves of a control experiment of graphene electrodes with the high set trigger at $10 G_0$ (The left panel is the closing process and the right panel is the opening process).

Reviewer 4:

Comments: The authors have responded satisfactorily to all the questions raised in my previous report. I am happy to recommend publication of the manuscript in its present form. I would like to note a typo in figure S3: the vertical axis should be “Conductance / $\log(G/G_0)$ ”.

Reply:

Thank reviewer for pointing this error. We are sorry for our mistake, and revise the Supplementary Figure 3 as follow:

Supplementary Figure 3 | The individual conductance-displacement curves of a control experiment of graphene electrodes with the high set trigger at $10 G_0$ (The left panel is the closing process and the right panel is the opening process).